# WHEN, WHY AND HOW MUCH?
# ADAPTIVE LEARNING RATE SCHEDULING BY REFINEMENT

## ABSTRACT

In this paper, we present a refined study of learning rate schedules for stochastic gradient descent (SGD). In contrast to most prior works that study the convergence of the average iterate, we study the last iterate, which is what most people use in practice. Furthermore, we break away from the tradition of replacing the gradients with crude upper bounds, which allows us to obtain a *problem-adaptive* learning rate schedule. Our method is the first systematic approach to *automatically* yield learning rate warm-up and rapid learning rate annealing near the end of training. In cases where gradient norm information is not available, our theory predicts that the best choice is the linear-decay schedule that sets the stepsize proportionally to $1-t/T$, where $t$ is the current iteration and $T$ is the total number of steps. Our final theoretical result is an extension of our methodology to coordinate-wise methods. We perform the most comprehensive evaluation of learning rate schedules to date, evaluating across 10 diverse deep learning problems, a series of LLMs, and a suite of logistic regression problems. We validate that overall, the linear-decay schedule outperforms all commonly used default schedules including cosine annealing, and that our schedule refinement method gives further improvements.

## 1 INTRODUCTION

For minimizing a function $f$, Stochastic Gradient Descent (SGD) updates the iterate $x_t$ at step $t$ via:

$$x_{t+1} = x_t - \eta_t g_t,$$

where $g_t$ is a (possibly stochastic) sub-gradient at $x_t$, and $\eta_t$ is the learning rate (LR) at time $t$. Choosing a sequence of $\eta_t$ for steps $t = 1, \ldots, T$ is a core problem in optimization.

The learning rate sequence for an optimizer is typically decomposed into two parts: the *baseline* learning rate, indicating the maximum LR to use, and a *schedule*, a sequence that multiplies the baseline LR to give the LR sequence. In this work we focus exclusively on the problem of scheduling. Choosing the right learning rate schedule for best performance is difficult; standard practice is to perform a hyper-parameter sweep over a set of standardized schedules (Wu et al., 2020).

Setting $\eta_t$ from theory is difficult due to a multitude of potential problem assumptions and the wildly varying schedules that arise from these assumptions. For instance, $\eta_t \propto 1/\sqrt{t}$, $\eta_t \propto 1/t$ and constant schedules $\eta_t = \eta$ are all dictated by three common but different sets of assumptions. Unfortunately, all three work suboptimally in practice for deep learning (Section 3.1), and are unpopular in the community (Ge et al., 2019). In this work we focus on two causes for this theory-practice gap:

1. Most theory analyzes the average iterate (Polyak, 1990; Ruppert, 1988) $\hat{x}_T = \frac{1}{T}\sum_{t=1}^{T} x_t$ or a randomly sampled iterate. However, in practice the last iterate $x_T$ is used.
2. Existing theory for the last iterate often uses crude constant bounds on the gradient norms or curvature. Our new tighter bound involves the entire gradient norm sequence instead, allowing for *problem-adaptive* LR schedules.

Our method is a "refinement" method: it uses a prior training run to produce an improved schedule to use in future runs. The practical variant of our schedule refinement method for SGD is given in Algorithm 1. Given a sequence of gradient norms produced by a prior run, it outputs a new schedule

---

**Algorithm 1** Schedule Refinement for SGD

---

1: **Input:** $G = \|g_t\|$ sequence of length $T$, smoothing hyper-parameter $\tau > 0$
2: $\hat{G} = \text{median\_filter}(G, \text{filter\_width} = \tau T, \text{padding} = (\text{nearest}, \text{reflect}))$
3: Define $w_t = \hat{G}_t^{-2}$
4: **for** $t = 1$ **to** $T$ **do**
5:

$$\eta_t = w_t \sum_{p=t+1}^{T} w_p$$

6: **end for**
7: Return normalized schedule $\eta/\max(\eta)$

---

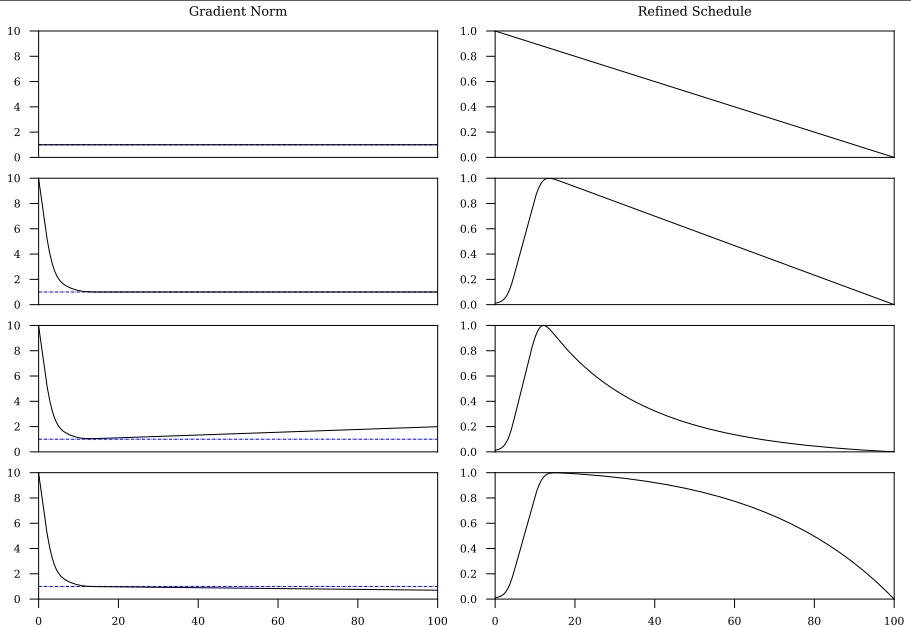

Figure 1: Example gradient norm sequences and the resulting refined schedules given by Algorithm 1. Blue line at $y = 1$ shown for reference. Percentage of runtime on the x-axis.

that is adaptive to the structure of the problem. Mathematically, it is minimizing a novel bound we derive on the function value $f(x_T)$ of the final iterate (Section 2.1). The learning rate at each time-step involves a sum of inverse-squared gradient norms from *future* time-steps, a major departure from previous approaches to scheduling.

Our analytical approach also departs from previous approaches by generalizing beyond SGD. Prior analyses of learning rate schedules often rely on the particular form of the SGD iterates to drive the calculations (Jain et al., 2019; Zamani & Glineur, 2023). Our approach is instead a broad technique that provides learning rate schedules for *any* base optimization algorithm. On a technical level, we design a step-size schedule that converts any algorithm which obtains a vanishing regret into one that ensures a last-iterate guarantee. This means that our refinement technique can provide theoretically-motivated schedules for popular base optimization algorithms like Adam.

We make contributions in both theory and practice. In theory, we provide a general analysis of learning rate schedules for arbitrary optimization algorithms. This recovers the optimal convergence rates for SGD, and also produces refined schedules customized for any task and optimizer. In practice, we show that warm-up followed by linear decay almost always matches or outperforms other common schedules, including cosine decay. Our refined schedules provide further improvements, suggesting that our theory provides guidance even for non-convex neural networks. Our refined schedules exhibit both warmup and nearly-linear decay (Figure 1). Zamani & Glineur (2023) have very recently shown last-iterate guarantees for linear decay, but to our knowledge this is the first time that warmup has arisen *directly from theory* rather than as an empirical heuristic (Goya et al., 2017).

## 1.1 NOTATION

$f \colon \mathbb{R}^d \to \mathbb{R}$ is a convex objective. $x_1, \ldots, x_T$ and $z_1, \ldots, z_T$ are random vectors in $\mathbb{R}^d$ with $x_1 = z_1$, and $\Delta_t \triangleq z_{t+1} - z_t$. $\Delta_t$ will indicate "baseline" updates before applying a schedule, and $x_t$ will be iterates after applying the schedule. $g_1, \ldots, g_T$ are random vectors in $\mathbb{R}^d$ satisfying $\mathbb{E}[g_t | x_1, \ldots, x_t] \in \partial f(x_t)$ ($\mathbb{E}[g_t] = \nabla f(x_t)$ when $f$ is differentiable). $G^2$ is a bound on $\mathbb{E}[\max_t \|g_t\|^2]$. $w_1, \ldots, w_T$ indicate non-negative random variables in $\mathbb{R}$ such that $g_t$ and $w_t$ are independent given $x_1, \ldots, x_t$. We define $w_{a:b} \triangleq \sum_{t=a}^{b} w_t$. If $a > b$, then $w_{a:b} \triangleq 0$. $u$ denotes an arbitrary element of $\mathbb{R}^d$; typically one should consider the case $u \in \arg\min f$. $D \triangleq \|x_1 - u\|$ is the "distance-to-solution" term, and $f_\star \triangleq \inf_u f(u)$.

## 2 MAIN ANALYTICAL RESULT

Our general result is Theorem 1. The regret of sequence $z_1, \ldots, z_T$ is defined as $\sum_{t=1}^{T} \langle g_t, z_t - u \rangle$. Assuming the regret is bounded, we convert into a sequence of iterates $x_1, \ldots, x_T$ with a bound on $f(x_T) - f(u)$. Thus, Theorem 1 can be viewed as another addition to the family of reductions from stochastic optimization to regret bounds (Cesa-Bianchi et al., 2004; Cutkosky, 2019). The proof can be found in Appendix C.

**Theorem 1** *Suppose $z_1, \ldots, z_T$ is some arbitrary sequence of vectors. Let $w_1, \ldots, w_T$ be an arbitrary sequence of non-negative numbers. Recall that we define $\Delta_t = z_{t+1} - z_t$ and $x_1 = z_1$. For $t \geq 1$, suppose $x_{t+1}$ satisfies:*

$$x_{t+1} = x_t + \frac{w_{t+1:T}}{w_{1:T}} \Delta_t,$$

*then for any $u$:*

$$\mathbb{E}[f(x_T) - f(u)] \leq \mathbb{E}\left[ \sum_{t=1}^{T} \frac{1}{w_{1:T}} \langle w_t \cdot g_t, z_t - u \rangle \right].$$

Let us take a moment to consider the implications of this Theorem in the simplest setting of $w_t = 1$ for all $t$. In this case, it is well-known that by setting $\Delta_t = -\eta g_t$ for $\eta = \frac{D}{G\sqrt{T}}$, one guarantees $\sum_{t=1}^{T} \langle g_t, z_t - u \rangle \leq DG\sqrt{T}$. Thus, we immediately obtain the following important corollary:

**Corollary 2** *Set $x_{t+1} = x_t - \eta_t g_t$ with $\eta_t = \frac{D}{G\sqrt{T}} \left(1 - \frac{t}{T}\right)$. Then:*

$$\mathbb{E}[f(x_T) - f(u)] \leq \frac{DG}{\sqrt{T}}.$$

**Proof** With $w_t = 1$ for all $t$ and $\Delta_t = -\frac{D}{G\sqrt{T}} g_t$, classical online gradient descent analysis (Zinkevich (2003)) yields $\sum_{t=1}^{T} \langle w_t g_t, z_t - u \rangle \leq DG\sqrt{T}$. The result now follows from Theorem 1. ∎

The sequence $x_t$ suggested by Corollary 2 is simply stochastic gradient descent ($x_{t+1} = x_t - \eta_t g_t$) equipped with a *linear decay learning rate schedule*:

$$\eta_t = \frac{D}{G\sqrt{T}} \left(1 - \frac{t}{T}\right). \tag{1}$$

Linear decay emulates the effects of iterate averaging, as the contribution from each gradient to the returned point is approximately the same as it would be in an average: the gradient $g_{T/2}$ appears in half the points in the average, and so its weight is halved, the gradient $g_t$ appears in $T - t$ out of $T$ points and so its weight is $1 - t/T$.

This bound has a significantly better constant than previous schedules for last-iterate convergence of SGD in this setting (Jain et al., 2019), and matches recent work by Zamani & Glineur (2023), who were the first to show that this schedule is actually optimal for gradient descent for the Convex $G$-Lipschitz complexity class. Our regret analysis recovers their result when specialized to SGD.

In practice, the linear decay schedule is employed not only with SGD but also with a diverse panoply of other optimization algorithms. Our Theorem 1 suggests a theoretical basis for this approach: so

long as the underlying optimization method ensures a regret bound, the linear decay schedule will provide a last-iterate guarantee. Note that we do not require the regret bound to be analytically proven (as is the case for e.g. SGD (Zinkevich, 2003), AdaGrad (Duchi et al., 2011; Streeter & McMahan, 2010) or AMSGrad (Reddi et al., 2018)); it suffices for the regret bound to hold in practice (as may hold for Adam (Kingma & Ba, 2015) or AdamW (Loshchilov & Hutter, 2017)).

## 2.1 Optimizing the bound for data-dependent schedules

We have now seen that setting $w_t = 1$ for all $t$ recovers the linear decay schedule, and can obtain the worst-case optimal convergence rates. However, optimizing for the worst case usually yields overly pessimistic behavior. In this section, we build more adaptive schedules that obtain better results on real data. To do this, we simply choose $w_t$ so as to optimize the bound in Theorem 1.

We focus on the particular case of SGD by setting $x_{t+1} = x_t - \eta_t g_t$ with $\eta_t = \frac{w_t w_{t+1:T}}{w_{1:T}}$. In the notation of Theorem 1, this corresponds to $\Delta_t = -w_t g_t$. For this case, we have the following result, with proof in Appendix D.

**Theorem 3** *Suppose that $x_{t+1} = x_t - \eta_t g_t$ with $\eta_t = \frac{w_t w_{t+1:T}}{w_{1:T}}$. Then we have:*

$$\mathbb{E}[f(x_T) - f(u)] \leq \mathbb{E}\left[\frac{1}{2 \cdot w_{1:T}}\left(D^2 + \sum_{t=1}^{T} w_t^2 \|g_t\|^2\right)\right]. \tag{2}$$

*Moreover, for a fixed sequence $\|g_1\|^2, \ldots, \|g_T\|^2$, the value of $\frac{1}{2 \cdot w_{1:T}}(D^2 + \sum_{t=1}^{T} w_t^2 \|g_t\|^2)$ is minimized by setting:*

$$w_t = \|g_t\|^{-2} \frac{D}{\sqrt{\sum_{p=1}^{T} \|g_p\|^{-2}}}.$$

Theorem 3 suggests that if we knew the sequence of gradient norms ahead of time, then we could optimize the weights $w_t$ (and therefore the learning rate schedule $\eta_t$) by setting $w_t \propto \|g_t\|^{-2}$. This yields a simple practical approach for *refining* a learning rate schedule based on empirical observations. First, perform one run using a baseline schedule to observe the sequence of gradient norms. Then, use these norms to compute an optimal schedule via Theorem 3. The constant factor $D = \|x_1 - u\|$ appearing in the value for $w_t$ plays the role of the "scale factor" typically applied to learning rate schedules. A line of recent work has shown that this quantity can be efficiently estimated online without significant downsides (Mcmahan & Streeter, 2012; Orabona & Pál, 2016; Cutkosky & Orabona, 2018; Mhammedi & Koolen, 2020; Zhang et al., 2022; Carmon & Hinder, 2022; Mishchenko & Defazio, 2023; Khaled et al., 2023; Cutkosky et al., 2023).

There are some subtleties here. Allowing $w_t$ to depend on random variable $g_t$ breaks the independence assumption between $w_t$ and $g_t$. Further, the act of changing the schedule from a baseline to a refined data-dependent schedule will change the gradient norm sequence, which may in turn indicate that a substantially different schedule would have been optimal. Our approach relies on the practical assumption that the gradient norm sequence does not change significantly after refinement. To encourage this, Algorithm 1 applies a median smoothing filter to the gradient norms before refining.

Finally, Theorem 3 provides an analysis that recovers schedules for SGD. However, practitioners commonly use optimization methods with more complicated *per-coordinate* updates or other pre-conditioning schemes such as Adam. In Appendix F we provide an alternative version of Theorem 3 that applies to such algorithms (Theorem 8). This result enables us to develop schedules tailored to any optimization algorithm. However, actually building this schedule in practice may require inspecting the algorithm's internal state, which may be difficult or inefficient. For per-coordinate algorithms like Adam, we suggest simply setting $w_t \propto 1/\|g_t\|_1$ as an approximation (see Section 3).

Figure 2.1 gives the Refined schedules on a set of standard benchmark machine learning problems when initially trained using linear decay schedules to provide the gradient norm sequences. Full details of the models and training setup for each problem is provided in the Appendix. Gradient $\ell_2$ norms are shown for SGD trained problems (ImageNet, RCNN) and $\ell_1$ norms for the Adam trained problems, which also use inverse-$\ell_1$ norm weights for refinement (see Section 3). Our single hyper-parameter $\tau = 0.3$ was tuned so that the resulting schedules obtained a balance between smoothness and capturing structure. We recommend setting this parameter *by eye* rather than by grid search.

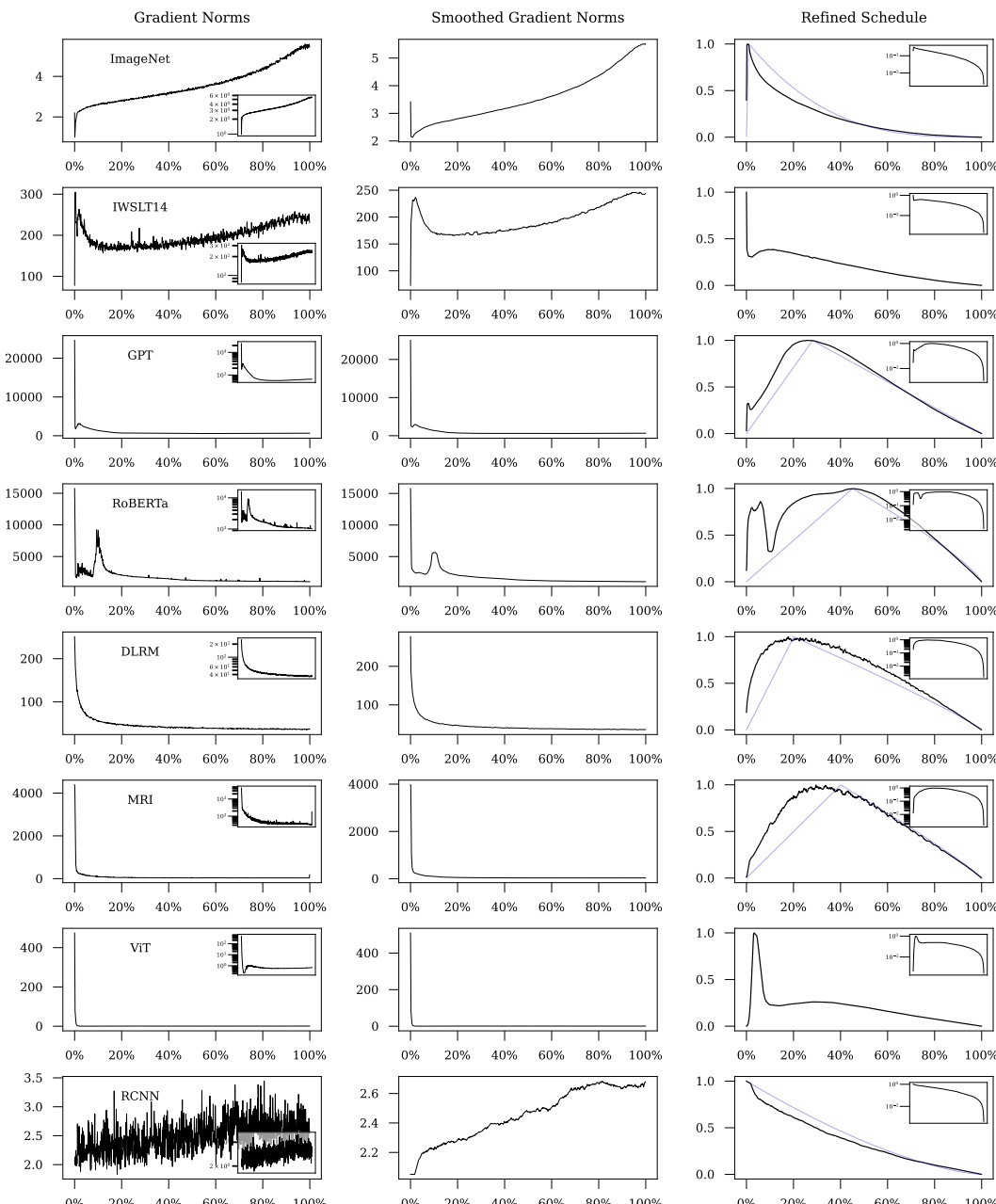

Figure 2: Gradient Norm sequences and the resulting Refined schedules, generated using an initial linear decay schedule with warmup for the initial run. Log scale views are inset for scale reference. Fitted polynomial decay schedules are overlayed in blue.

## 3 EXPERIMENTS

To validate the effectiveness of our method on convex problems, we performed a comparison across 8 commonly used classification problems from the LIBSVM repository, with separable problems excluded (see Section A). We used a logistic regression loss together with the Adam optimizer with $\beta = (0.9, 0.95)$, with batch size 16 and trained for 100 epochs. Table 3 demonstrates that both the linear decay schedule and our refinement schedule consistently either match or out-perform the cosine schedule. The linear decay schedule matches the cosine schedule on every problem (up to statistical significance), and out-performs it on two problems. The Refined schedule further out-performs the linear decay schedule, either matching of exceeding it across the board.

Table 1: Logistic Regression Experiments (Train Error Rate %).

| Problem | Stepwise | Cosine | Linear | Refined $\|g\|_2^2$ | Refined $\|g\|_1$ | Refined |
|---|---|---|---|---|---|---|
| Aloi | 13.38 ±0.05 | 12.69 ±0.05 | 12.76 ±0.06 | **11.75** ±0.01 | 12.24 ±0.04 | 12.30 ±0.05 |
| Glass | 31.39 ±0.16 | 30.82 ±0.22 | 30.72 ±0.18 | 30.05 ±0.44 | **29.62** ±0.37 | 30 ±0.41 |
| Iris | **1.39** ±0.000 | **1.39** ±0.000 | **1.39** ±0.000 | 1.46 ±0.07 | 1.46 ±0.07 | 1.46 ±0.07 |
| Letter | 22.24 ±0.008 | **22.24** ±0.01 | 22.20 ±0.02 | 22.23 ±0.03 | 22.20 ±0.03 | 22.20 ±0.03 |
| Pendigits | 4.70 ±0.02 | 4.67 ±0.01 | **4.62** ±0.03 | **4.56** ±0.02 | 4.58 ±0.02 | **4.56** ±0.04 |
| Sensorless | 11.84 ±0.09 | 11.30 ±0.09 | 11.29 ±0.09 | 10.71 ±0.08 | **10.08** ±0.05 | 10.11 ±0.05 |
| Vehicle | 18.83 ±0.09 | 18.49 ±0.05 | 18.55 ±0.06 | **18.21** ±0.12 | 18.19 ±0.08 | **18.21** ±0.1 |
| Vowel | 23.43 ±0.08 | 22.99 ±0.09 | 22.94 ±0.10 | **22.48** ±0.12 | 22.44 ±0.08 | 22.41 ±0.08 |

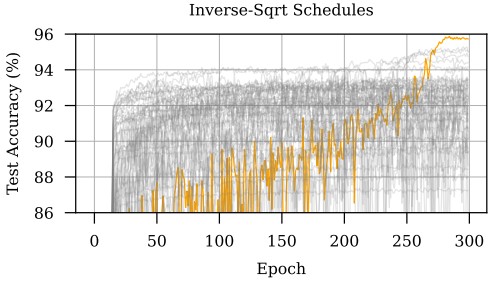 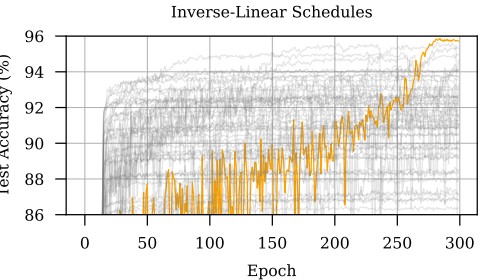

Figure 3: Training curves on CIFAR-10 for a sweep over inverse-sqrt and inverse-linear hyper-parameters. A linear-decay schedule baseline is shown in orange. All combinations are out-performed by the linear-decay schedule.

Rather than using the Adam specific weighting for deriving the Refined schedules, it is often convenient to use other norms, particularly if existing logs are available using these other norms. In Table 3 we present results using both the $\ell_1$ norm and $\ell_2$ norm-squared weighting. Both are shown to maintain the advantages of the refinement technique, out-performing the non-refined schedules consistently. Given the ease of logging $\ell_1$ norms compared to the internal Adam state, we advocate for weights to be the inverse of the $\ell_1$ norm (no squaring) when using Adam.

### 3.1 DEEP LEARNING EXPERIMENTS

Classical any-time learning rates schedules such as $1/\sqrt{t}$ and $1/t$ are commonly used in theoretical analysis of optimization methods, yet they are rarely used by practitioners. To illustrate why, in Figure 3.1, we sweep over the learning rate $\eta$ and offset $\beta$ for schedules of the form:

$$\eta\frac{\beta}{\beta+t} \quad \text{and} \quad \eta\frac{\sqrt{\beta}}{\sqrt{\beta+t}}.$$

A 5% duration learning rate warmup was also included in each case. Even with 60 hyper-parameter combinations tried for each family, neither are able to match the performance of the linear-decay schedule. For our deep learning experiments we implemented two commonly used practical variants of these schedules, since sweeping the $\beta$ hyper-parameter is computationally impractical. The vanilla version uses $\beta = 1$, and the *offset* version uses $\beta$ set to match the duration of the warmup period (a common setting due to its implementation in FairSeq).

### 3.2 LARGE-SCALE SCHEDULE BENCHMARK

We performed a large-scale comparison of schedules across common deep learning benchmarks:

**CIFAR10** For a small-scale image-classification problem, we chose CIFAR10 (Krizhevsky, 2009). A high-performance Wide ResNet architecture was used (Zagoruyko & Komodakis, 2016). Test error percentage is reported.

**CIFAR100** A more realistic yet still small scale benchmark. We used a compact DenseNet architecture (Huang et al., 2017) to increase the diversity of model architectures tested. Test error percentage is reported.

**ImageNet** We used a ResNet-50 architecture (He et al., 2016) for the ImageNet (Russakovsky et al., 2015) classification task. Note that we used a standard (non-SOTA) data-augmentation pipeline following other recent comparisons in the optimization literature. Test error percentage is reported.

**IWSLT14** A small-scale German-English translation task (Cettolo et al., 2014) using a LSTM model (Wiseman & Rush, 2016). Test Perplexity is reported.

Table 2: Classical Schedule Comparison Against the Linear Schedule (lower = better).

| Problem | Flat | 1/t | 1/sqrt | Offset 1/t | Offset 1/sqrt | Linear |
|---|---|---|---|---|---|---|
| CIFAR10 | 8.04 ±.13 | 5.42 ±.28 | 6.37 ±.41 | 9.23 ±.08 | 6.58 ±.06 | **4.35** ±.05 |
| CIFAR100 | 30.43 ±.20 | 26.58 ±.11 | 29.09 ±.16 | 32.80 ±.07 | 27.62 ±.13 | **22.11** ±.08 |
| ImageNet | 33.00 ±.14 | 26.48 ±.06 | 28.35 ±.05 | 47.94 ±.08 | 27.34 ±.06 | **23.11** ±.07 |
| IWSLT14 | 8.07 ±.02 | 7.62 ±.01 | 7.52 ±.01 | 12.89 ±.06 | 8.48 ±.01 | **7.10** ±.01 |
| GPT | 20.20 ±.000 | 18.99 ±.04 | 19.48 ±.02 | 27.85 ±.05 | 22.88 ±.006 | **18.60** ±.02 |
| RoBERTa | 4.52 ±.005 | 4.25 ±.007 | 4.33 ±.01 | 5.33 ±.02 | 5.15 ±.02 | **3.94** ±.007 |
| DLRM | **20.95** ±.006 | 47.59 ±6.45 | 45.99 ±5.98 | **20.94** ±.007 | 20.99 ±.009 | **20.94** ±.006 |
| MRI | **9.00** ±.04 | 8.91 ±.01 | **8.98** ±.04 | 9.53 ±.08 | 9.16 ±.05 | **8.88** ±.02 |
| ViT | 30.11 ±.27 | 28.36 ±.40 | 28.53 ±.15 | 73.84 ±6.08 | 50.36 ±12.39 | **24.82** ±.31 |
| RCNN | 65.43 ±.12 | 63.38 ±.05 | 64.13 ±.10 | 79.32 ±.07 | 69.25 ±.07 | **60.98** ±.02 |

Table 3: Modern Schedule Comparison (lower = better).

| Problem | Stepwise | Cosine | Linear | Refinement |
|---|---|---|---|---|
| CIFAR10 | 4.53 ±.03 | **4.27** ±.04 | **4.35** ±.05 | - |
| CIFAR100 | 22.78 ±.10 | 22.59 ±.09 | **22.11** ±.08 | - |
| ImageNet | 23.51 ±.07 | **23.10** ±.06 | **23.11** ±.07 | **23.12** ±0.03 |
| IWSLT14 | 7.43 ±.01 | 7.17 ±.009 | 7.10 ±.01 | **6.92** ±.03 |
| GPT | 19.70 ±.03 | 18.65 ±.02 | 18.60 ±.02 | **18.29** ±.005 |
| RoBERTa | 4.36 ±.01 | 4.07 ±.000 | 3.94 ±.007 | **3.86** ±.005 |
| DLRM | 20.95 ±.008 | **20.94** ±.005 | **20.94** ±.006 | **20.94** ±.009 |
| MRI | 8.97 ±.02 | 8.90 ±.03 | 8.88 ±.02 | **8.85** ±.01 |
| ViT | 26.27 ±.33 | **24.56** ±.15 | 24.82 ±.31 | 25.53 ±.16 |
| RCNN | 61.76 ±.06 | 61.00 ±.04 | **60.98** ±.02 | 61.88 ±.02 |

**GPT** A modern small (162M parameter) GPT-style auto-regressive transformer model (Radford et al., 2019) trained on the large Book-Wiki corpus (Zhu et al., 2015). Test Perplexity is reported.

**RoBERTa** A masked autoencoder variant (Liu et al., 2019) also trained on the Book-Wiki corpus. Test Perplexity is reported.

**ViT** A high-performance ImageNet classifier using a Vision Transformer (Dosovitskiy et al., 2021). In contrast to our ResNet-50 experiments, here we use modern data-augmentation, following standard practice for ViT training. Test error percentage is reported.

**DLRM** A modern recommendation system engine (Naumov et al., 2019) trained on the open Criteo Kaggle Display Advertising dataset. Test Accuracy is reported.

**MRI** A large stacked U-Net architecture (Sriram et al., 2020) trained on the fastMRI dataset (Zbontar et al., 2018), an image-to-image regression task from the medical imaging domain. Test set metric $100 \cdot (1 - SSIM)$ is reported.

**RCNN** The object detection method Faster-RCNN (Ren et al., 2015) trained on the COCO 2017 dataset, using a ResNet-50 ImageNet pretrained backbone. $100 - $ box AP is reported.

For each problem, we first performed a sweep of learning rates on a grid $[10^i, 2 \times 10^i, 5 \times 10^i]$ for varying $i$, separately for each schedule. We then ran multiple seeds using the best learning-rate from the grid search. Mean and standard error of the mean was tabulated for each schedule. The best result for each method, up to a statistical significance level of 0.05 using a paired two-sample t-test, is highlighted in bold. Specific details of the hyper-parameters used for each problem are given in Appendix H. All non-adaptive schedules included a fixed learning-rate warmup, with length following standard practices for the problem. Our step-wise schedule uses a 30-60-90 percent tenthing, following standard practices from ImageNet training.

Tables 2 & 3 show the results. We break the schedules into two categories, classical and modern. The modern schedules consistently outperform the classical schedules, often by large margins. Although this is common folk-law in deep learning, we are not aware of any existing large-scale experiments establishing this. Our comparison of modern schedules shows a clear hierarchy among the schedules. The Stepwise schedule is dominated by the Cosine schedule, and the Linear schedule matches or outperforms the Cosine schedule on all problems except ViT. The refined schedule further outperforms the Linear schedule on 5 of the 10 problems, but shows mixed results on ViT and RCNN. The refinement process produced degenerate schedules that fail on the CIFAR problems, we discuss this in Appendix A.

Table 4: CIFAR10 Training Time Ablations (Test Error %).

| Schedule | 1 Epoch | 5 Epochs | 15 Epochs | 30 Epochs |
|---|---|---|---|---|
| Cosine | $35.80 \pm 0.27$ | $15.07 \pm 0.11$ | $7.99 \pm 0.06$ | $\mathbf{6.08} \pm 0.03$ |
| Linear Decay | $\mathbf{33.84} \pm 0.17$ | $\mathbf{14.36} \pm 0.07$ | $\mathbf{7.65} \pm 0.07$ | $6.12 \pm 0.08$ |
| Refined | $\mathbf{34.04} \pm 0.19$ | $\mathbf{14.59} \pm 0.08$ | $7.88 \pm 0.04$ | $6.24 \pm 0.06$ |

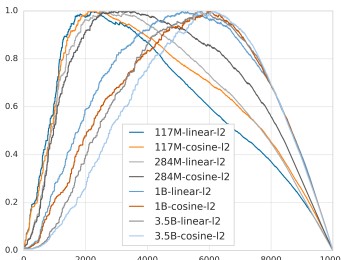 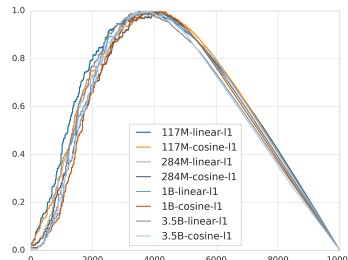

Figure 4: Learning rate schedules learned on vanilla Transformer-based LLM trained on C4 dataset. Left: schedules obtained when setting $w_t \propto \|g_t\|^{-2}$. Right: schedules when $w_t \propto 1/\|g_t\|_1$. We find that $\ell_1$ norm is much more consistent across model sizes and baseline schedules. Thus, we used $w_t \propto 1/\|g_t\|_1$ weighting from linear baselines to obtained Refined schedule mentioned in Table 5.

### 3.3 COSINE SCHEDULE ABLATIONS

To further investigate the failure modes of the cosine schedule, we ran a series of shorter duration training runs on CIFAR10. Our premise was that the cosine schedule is heavily over-fit to long training duration computer vision problems. As shown in Table 4, the cosine schedule begins to under-perform the linear decay schedule and the refined schedules when training for less than 30 epochs. In contrast, while the refined schedule also under-performs for longer duration training it has no statistically significant differences from the linear decay schedule for shorter duration training, where they both perform consistently better than the cosine schedule.

### 3.4 LARGE LANGUAGE MODEL SIZE ABLATIONS

In addition to the results above, we validate our insights on Language Models by performing an additional set of experiments directly comparing to the setting explored in Chinchilla (Hoffmann et al., 2022) where we train a vanilla Transformer model on the C4 dataset (Raffel et al., 2020) for 10k steps with a token batch size of around 524k ($2^{19}$). Starting with Hoffmann et al. (2022), most recent LLMs employ the AdamW optimizer with Linear Warmup and Cosine Decay. We perform a head-to-head comparison of Cosine decay, Linear decay and Refined schedules (the latter is refined from Linear decay). As shown in Table 5, contrary to popular wisdom, Linear decay performs better than Cosine decay across all model sizes. The Refined schedule outperforms Linear Decay for all but the 3.5B sized model.

## 4 DISCUSSION

Gradient sequences that slowly decrease or slowly increase after stabilizing are often observed in practice (Figure 2.1), and the resulting schedules that our framework produces show interesting behavior, resembling *polynomial-decay* schedules with exponents $p$:

$$\eta_t \propto \left(1 - \frac{t}{T}\right)^p. \tag{3}$$

Schedules of this form are already in use, with $p$ typically tuned as a hyper-parameter. From Figure 1, we see that $p < 1$ should be used when the gradient norm is decreasing over time, $p > 1$ when the gradient norm is increasing, and $p = 1$ when the gradient sequence is flat. We use the term *polynomial decay* for the schedule given in Equation 3, which its name in both PyTorch and Tensorflow, although in the literature schedules of the form $1/t^\alpha$ are also sometimes referred to as polynomial schedules. From Figure 2.1, we can see that many of our test problems have refined schedules that are well-approximated by a polynomial decay sequences, shown in blue.

Table 5: LLM Ablations (C4 validation loss).

| Schedule | 117M | 284M | 1B | 3.5B |
|---|---|---|---|---|
| Cosine | 3.089 | 2.891 | 2.729 | 2.631 |
| Linear Decay | 3.087 | 2.888 | 2.725 | **2.625** |
| Refined | **3.075** | **2.884** | **2.722** | 2.634 |

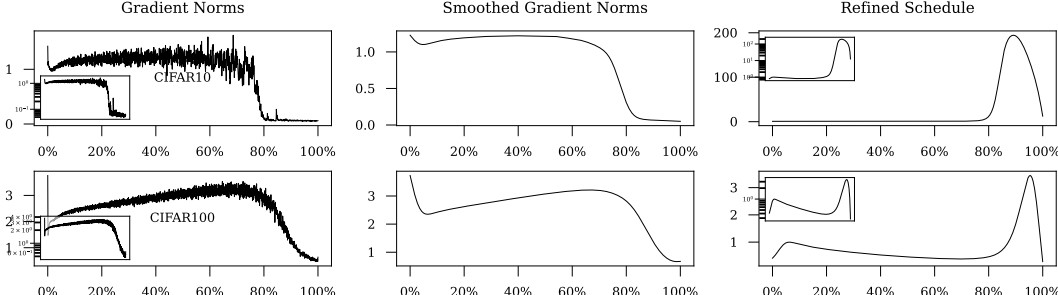

Figure 5: Limitations of refinement: if model over-fits, our method produces degenerate schedules.

## 5 RELATED WORK

The standard schedule for strongly-convex stochastic optimization is $\eta_t \propto 1/t$. This optimal when appropriate averaging is used (Shamir & Zhang, 2013; Lacoste-Julien et al., 2012; Rakhlin et al., 2012). For non-strongly convex optimization, theory typically recommends a constant schedule.

Learning rate schedules used in applications usually fall within a few standard classes supported by major frameworks. To survey the use of schedulers in open source software, we performed a GitHub search on each of the schedulers in the PyTorch Library. Table 6 (Appendix) shows that the polynomial decay schedule, which includes the linear decay schedule as a special case, is currently the least popular scheduler in PyTorch. Step-wise schedules are by far the most popular.

Our approach is motivated by the remarkable work of Pan et al. (2022), who show that for the last iterate of quadratic problems, minimizing an upper bound can yield improved schedules. However their work requires knowledge of the full Eigenspectrum of the Hessian, making it impractical to use. Our theory relies the sequence of expected gradient norms, a more tractable quantity.

Ours is not the first reduction from regret bounds to a last-iterate guarantee: Cutkosky (2019) suggests evaluating gradients at running averages to achieve the same convergence rate as Theorem 1. However, our method is directly interpretable in terms of learning rate schedules to yield immediate practical guidance while the previous technique is a more opaque transformation of the base algorithm. Nevertheless, understanding the differences is a valuable future direction.

There has been significant work on learning rate adaptation using gradient or loss information. The AdaGradNorm-type schedule is backed by strong theory (Wu et al., 2020; Streeter & McMahan, 2010; Duchi et al., 2011; Ward et al., 2019; Li & Orabona, 2019). They suggest $\eta_t \propto 1/\sqrt{\sum_{t=1}^{T} \|g_t\|^2}$, but it is unclear how to apply these schedules to other optimizers such as Adam.

Methods based on hyper-gradients (i.e. gradients of the learning rate) are another well-studied class of adaptive method. They use recent gradient evaluations and other local information to update the learning rate (Almeida et al., 1999; Franceschi et al., 2017; Bengio, 2000; Domke, 2012; Pedregosa, 2016; Baydin et al., 2018; Feurer & Hutter, 2019; Donini et al., 2020; Chandra et al., 2022). Alternatively, hyper-gradients can be used to signal when to decrease the learning rate (Pflug, 1983; 1988; Lang et al., 2019; Zhang et al., 2020). Methods based on loss minimization choose a learning rate that greedily minimizes the test or train loss (Xu et al., 2019; Jin et al., 2021; Kim et al., 2021).

### CONCLUSION

Our experimental results strongly support the use of the linear decay schedule as a default baseline for deep learning optimization. Applying refinement on top of an initial linear decay run gives a schedule showing improvements on 5 out of 10 deep learning problems and 5 of 8 convex problems. We develop extensive theory supporting the linear decay schedule and our novel refinement technique, establishing that their SOTA performance is well-supported both in theory and practice.

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

## A    LIMITATIONS

There are certain problems like CIFAR-10 and CIFAR-100 where we can clearly observe potential pitfalls of our approach (Figure 4) when our assumptions break down. On these problems $100\%$ train accuracy is reached near the end, driving the gradient norm sequence to zero. The resulting refined schedule rapidly increases the step size near the end of optimization, which if used results in training divergence. We suggest using a linear decay schedule instead on problems where the gradient norm drops significantly at the end.

## B    ALL-TAIL SUMMATION BOUND

In this section we develop some key Lemmas that underlie our results. The starting point of this development is the following result from Orabona (2020); Lin et al. (2016). We present the result and proof below for completeness, and then provide an improved version of the bound (Lemma 5) that is used to derive our analytical results.

### B.1    EXISTING BOUND

**Lemma 4** *Let $q_t \geq 0$, and let $\eta_t$ be a positive non-increasing sequence. Then:*

$$\eta_T q_T \leq \frac{1}{T} \sum_{t=1}^{T} \eta_t q_t + \sum_{k=1}^{T-1} \frac{1}{k(k+1)} \sum_{t=k+1}^{T} \eta_t \left(q_t - q_k\right),$$

**Proof**  We present below a version of his proof that's written to move the single inequality application tion in the whole proof to the last step. Define $S_k = \frac{1}{k} \sum_{t=T-k+1}^{T} \eta_t q_t$.

Then

$$\begin{aligned} kS_k &= (k+1)S_{k+1} - \eta_{T-k} q_{T-k} \\ &= ks_{k+1} + S_{k+1} - \eta_{T-k} q_{T-k} \\ &= ks_{k+1} + \frac{1}{k+1} \sum_{t=T-k}^{T} \left(\eta_t q_t - \eta_{T-k} q_{T-k}\right). \end{aligned}$$

Dividing through by $k$:

$$S_k = S_{k+1} + \frac{1}{k(k+1)} \sum_{t=T-k}^{T} \left(\eta_t q_t - \eta_{T-k} q_{T-k}\right).$$

Unrolling, we have:

$$S_1 = S_T + \sum_{k=1}^{T-1} \frac{1}{k(k+1)} \sum_{t=T-k}^{T} \left(\eta_t q_t - \eta_{T-k} q_{T-k}\right).$$

Now we use $S_1 = \eta_T q_T$. Note that the first entry in that sum is zero so we may shift the indexing to start at $t = T - k + 1$. Giving:

$$\begin{aligned} \eta_T q_T &= S_T + \sum_{k=1}^{T-1} \frac{1}{k(k+1)} \sum_{t=T-k+1}^{T} \left(\eta_t q_t - \eta_{T-k} q_{T-k}\right) \\ &\leq S_T + \sum_{k=1}^{T-1} \frac{1}{k(k+1)} \sum_{t=T-k+1}^{T} \eta_t \left(q_t - q_{T-k}\right). \end{aligned}$$

Where the final step uses the fact that $\eta_t$ is decreasing. Plugging in the definition of $S_T$, and substituting $k = T - k$ to simplify gives the result.  ∎

## B.2 IMPROVED EXPRESSION

**Lemma 5** *Let $q_t$ be any sequence, and let $w_t$ be a positive sequence. Then:*

$$q_T = \frac{1}{\sum_{t=1}^T w_t} \sum_{t=1}^T w_t q_t + \sum_{k=1}^{T-1} \frac{w_k}{\sum_{t=k+1}^T w_t} \left( \frac{1}{\sum_{t=k}^T w_t} \sum_{t=k}^T w_t \left( q_t - q_k \right) \right)$$

$$= \frac{1}{w_{1:T}} \sum_{t=1}^T w_t q_t + \sum_{k=1}^{T-1} \left( \frac{1}{w_{k+1:T}} - \frac{1}{w_{k:T}} \right) \sum_{t=k}^T w_t \left( q_t - q_k \right).$$

**Proof** Define

$$S_k = \frac{1}{\sum_{t=T-k+1}^T w_t} \sum_{t=T-k+1}^T w_t q_t.$$

Note that with this definition:

$$S_1 = \frac{1}{\sum_{t=T}^T w_t} \sum_{t=T}^T w_t q_t = q_T,$$

and $S_T$ is the full sum:

$$S_T = \frac{1}{\sum_{t=1}^T w_t} \sum_{t=1}^T w_t q_t.$$

The difference from the weighting used in the the Lemma above: we normalized by the sum of the step sizes rather than $k$. We get the following expansion:

$$\left( \sum_{t=T-k+1}^T w_t \right) S_k = \sum_{t=T-k+1}^T w_t q_t$$

$$= \sum_{t=T-k}^T w_t q_t - w_{T-k} q_{T-k}$$

$$= \left( \sum_{t=T-k}^T w_t \right) S_{k+1} - w_{T-k} q_{T-k}$$

$$= \left( \sum_{t=T-k+1}^T w_t \right) S_{k+1} + \left( w_{T-k} S_{k+1} - w_{T-k} q_{T-k} \right).$$

So dividing through by $\sum_{t=T-k+1}^T w_t$:

$$S_k = S_{k+1} + \frac{w_{T-k}}{\sum_{t=T-k+1}^T w_t} \left( S_{k+1} - q_{T-k} \right).$$

Unrolling

$$S_1 = S_T + \sum_{k=1}^{T-1} \frac{w_{T-k}}{\sum_{t=T-k+1}^T w_t} \left( S_{k+1} - q_{T-k} \right).$$

Note that, plugging in $S_{k+1}$:

$$S_{k+1} - q_{T-k} = \frac{1}{\sum_{t=T-k}^T w_t} \sum_{t=T-k}^T w_t q_t - q_{T-k}$$

$$= \frac{1}{\sum_{t=T-k}^T w_t} \sum_{t=T-k}^T w_t \left( q_t - q_{T-k} \right).$$

So we have:

$$q_T = \frac{1}{\sum_{t=1}^{T} w_t} \sum_{t=1}^{T} w_t q_t + \sum_{k=1}^{T-1} \frac{w_{T-k}}{\sum_{t=T-k+1}^{T} w_t} \left( \frac{1}{\sum_{t=T-k}^{T} w_t} \sum_{t=T-k}^{T} w_t \left( q_t - q_{T-k} \right) \right).$$

Finally, make the change of variables $k \to T - k$ and simply to yield the result. ∎

**Remark 6** *We will typically use this result by setting $q_t = f(x_t) - f_*$ in order to bound $q_T = f(x_T) - f_*$. By using a weighting sequence $w_t$ elsewhere, we are able to remove the $w_T$ weight from in front of the $q_T$ term (in contrast to Lemma 4). This is crucial, as we want to be able to analyze the situation in which $w_t$ drops extremely small near the end, and yet still bound $q_T = f(x_T) - f_*$, but if $q_T$ is weighted by $w_T$ we will get very loose bounds when $w_T$ is small. Notice also that we have an equality instead of an inequality, and we have not had to impose the requirement that $w$ be a non-increasing sequence.*

## C  PROOF OF THEOREM 1

Before proving Theorem 1, we need the following important Lemma:

**Lemma 7** *Suppose $z_1, \ldots, z_T$ is an arbitrary sequence of vectors. Let $w_1, \ldots, w_T$ be an arbitrary sequence of positive numbers. Define the sequence $x_1, \ldots, x_T$ recursively by $x_1 = z_1$ and:*

$$x_t = w_{t:T} \left( \frac{z_t}{w_{1:T}} + \sum_{p=1}^{t-1} x_p \left( \frac{1}{w_{p+1:T}} - \frac{1}{w_{p:T}} \right) \right).$$

*Suppose $g_t$ are random variables. with $\mathbb{E}[g_t | x_1, \ldots, x_t] \in \partial f(x_t)$ for some convex $f$. Then:*

$$\mathbb{E}[f(x_T) - f(u)] \le \mathbb{E}\left[ \frac{1}{w_{1:T}} \sum_{t=1}^{T} w_t \langle g_t, z_t - u \rangle \right].$$

**Proof** Let us set $q_t = f(x_t) - f(u)$. Then we have $\mathbb{E}[q_t] \le \mathbb{E}[\langle g_t, x_t - u \rangle]$ and $\mathbb{E}[q_t - q_k] \le \mathbb{E}[\langle g_t, x_t - x_k \rangle]$. Then, Theorem 5 implies:

$$\mathbb{E}[q_T] \le \mathbb{E}\left[ \frac{1}{w_{1:T}} \sum_{t=1}^{T} w_t \langle g_t, x_t - u \rangle + \sum_{k=1}^{T-1} \left( \frac{1}{w_{k+1:T}} - \frac{1}{w_{k:T}} \right) \sum_{t=k}^{T} w_t \langle g_t, x_t - x_k \rangle \right].$$

Now, let us find the coefficient of $\langle g_t, x_t \rangle$ in the above expression. This is:

$$\frac{w_t}{w_{1:T}} + \left[ \sum_{k=1}^{t} \left( \frac{1}{w_{k+1:T}} - \frac{1}{w_{k:T}} \right) w_t \right] - w_t \left( \frac{1}{w_{t+1:T}} - \frac{1}{w_{t:T}} \right)$$

$$= \frac{w_t}{w_{1:T}} + \sum_{k=1}^{t-1} \left( \frac{1}{w_{k+1:T}} - \frac{1}{w_{k:T}} \right) w_t$$

$$= \frac{w_t}{w_{t:T}}.$$

Next, for $p < t$, the coefficient of $\langle g_t, x_p \rangle$ is:

$$- \left( \frac{1}{w_{p+1:T}} - \frac{1}{w_{p:T}} \right) w_t.$$

And for $p > t$, the coefficient of $\langle g_t, x_p \rangle$ is zero. Finally the coefficient of $\langle g_t, u \rangle$ is $-\frac{w_t}{w_{1:T}}$.

Putting this all together, we can rearrange the expression as follows:

$$\mathbb{E}[q_T] \le \mathbb{E}\left[ \sum_{t=1}^{T} \left\langle g_t, \frac{w_t}{w_{t:T}} x_t - \left( \frac{w_t u}{w_{1:T}} + \sum_{p=1}^{t-1} w_t x_t \left( \frac{1}{w_{p+1:T}} - \frac{1}{w_{p:T}} \right) \right) \right\rangle \right]$$

$$= \mathbb{E}\left[ \sum_{t=1}^{T} \left\langle \frac{w_t}{w_{t:T}} g_t, x_t - w_{t:T} \left( \frac{u}{w_{1:T}} + \sum_{p=1}^{t-1} x_p \left( \frac{1}{w_{p+1:T}} - \frac{1}{w_{p:T}} \right) \right) \right\rangle \right].$$

Now, given an arbitrary sequence $z_1, \dots, z_T$, define $x_t$ recursively by:

$$x_t = w_{t:T}\left(\frac{z_t}{w_{1:T}} + \sum_{p=1}^{t-1} x_p\left(\frac{1}{w_{p+1:T}} - \frac{1}{w_{p:T}}\right)\right).$$

Then we have:

$$\mathbb{E}[q_T] \le \mathbb{E}\left[\sum_{t=1}^{T}\left\langle \frac{w_t}{w_{t:T}}g_t, x_t - w_{t:T}\left(\frac{u}{w_{1:T}} + \sum_{p=1}^{t-1} x_p\left(\frac{1}{w_{p+1:T}} - \frac{1}{w_{p:T}}\right)\right)\right\rangle\right]$$

$$= \mathbb{E}\left[\sum_{t=1}^{T}\frac{w_t}{w_{1:T}}\langle g_t, z_t - u\rangle\right].$$

∎

Now, we are finally ready to prove Theorem 1:

**Theorem 1** *Suppose $z_1, \dots, z_T$ is some arbitrary sequence of vectors. Let $w_1, \dots, w_T$ be an arbitrary sequence of non-negative numbers. Recall that we define $\Delta_t = z_{t+1} - z_t$ and $x_1 = z_1$. For $t \ge 1$, suppose $x_{t+1}$ satisfies:*

$$x_{t+1} = x_t + \frac{w_{t+1:T}}{w_{1:T}}\Delta_t,$$

*then for any $u$:*

$$\mathbb{E}[f(x_T) - f(u)] \le \mathbb{E}\left[\sum_{t=1}^{T}\frac{1}{w_{1:T}}\langle w_t \cdot g_t, z_t - u\rangle\right].$$

**Proof** Let's define $\hat{x}_1 = z_t$ and recursively set:

$$\hat{x}_t = w_{t:T}\left(\frac{z_t}{w_{1:T}} + \sum_{p=1}^{t-1} \hat{x}_p\left(\frac{1}{w_{p+1:T}} - \frac{1}{w_{p:T}}\right)\right).$$

Then, Lemma 7 shows that $\mathbb{E}[f(\hat{x}_T) - f(u)] \le \mathbb{E}\left[\frac{1}{w_{1:T}}\sum_{t=1}^{T} w_t\langle g_t, z_t - u\rangle\right]$. So, it suffices to show that $x_t = \hat{x}_t$ for all $t$. In turn, since $\hat{x}_1 = z_1 = x_1$, it suffices to show $\hat{x}_{t+1} - \hat{x}_t = \frac{w_{t+1:T}}{w_{1:T}}\Delta_t = x_{t+1} - x_t$ for all $t$.

To this end, let's do some calculation. First:

$$\hat{x}_t = \frac{w_t}{w_{t:T}}\hat{x}_t + \frac{w_{t+1:T}}{w_{t:T}}\hat{x}_t$$

$$= \frac{w_t}{w_{t:T}}\hat{x}_t + w_{t+t:T}\left(\frac{z_t}{w_{1:T}} + \sum_{p=1}^{t-1} \hat{x}_p\left(\frac{1}{w_{p+1:T}} - \frac{1}{w_{p:T}}\right)\right).$$

With this expression, we have:

$$\hat{x}_{t+1} - \hat{x}_t = \hat{x}_{t+1} - w_{t+1:T}\left(\frac{z_{t+1}}{w_{1:T}} + \sum_{p=1}^{t} x_p\left(\frac{1}{w_{p+1:T}} - \frac{1}{w_{p:T}}\right)\right) - \frac{w_t}{w_{t:T}}\hat{x}_t$$

$$= w_{t+1:T}\left(\frac{z_{t+1}}{w_{1:T}} + \sum_{p=1}^{t} \hat{x}_p\left(\frac{1}{w_{p+1:T}} - \frac{1}{w_{p:T}}\right)\right)$$

$$\quad - w_{t+1:T}\left(\frac{z_t}{w_{1:T}} + \sum_{p=1}^{t-1} x_p\left(\frac{1}{w_{p+1:T}} - \frac{1}{w_{p:T}}\right)\right) - \frac{w_t}{w_{t:T}}\hat{x}_t$$

$$= \frac{w_{t+1:T}(z_{t+1} - z_t)}{w_{1:T}} + w_{t+1:T}\hat{x}_t\left(\frac{1}{w_{t+1:T}} - \frac{1}{w_{t:T}}\right) - \frac{w_t}{w_{t:T}}\hat{x}_t$$

$$= \frac{w_{t+1:T}}{w_{1:T}}\Delta_t + \hat{x}_t\left(1 - \frac{w_{t+1:T} + w_t}{w_{t:T}}\right)$$

$$= \frac{w_{t+1:T}}{w_{1:T}}\Delta_t.$$

■

## D    PROOF OF THEOREM 3

**Theorem 3** *Suppose that $x_{t+1} = x_t - \eta_t g_t$ with $\eta_t = \frac{w_t w_{t+1:T}}{w_{1:T}}$. Then we have:*

$$\mathbb{E}[f(x_T) - f(u)] \leq \mathbb{E}\left[\frac{1}{2 \cdot w_{1:T}}\left(D^2 + \sum_{t=1}^{T} w_t^2 \|g_t\|^2\right)\right]. \tag{2}$$

*Moreover, for a fixed sequence $\|g_1\|^2, \ldots, \|g_T\|^2$, the value of $\frac{1}{2 \cdot w_{1:T}}(D^2 + \sum_{t=1}^{T} w_t^2 \|g_t\|^2)$ is minimized by setting:*

$$w_t = \|g_t\|^{-2} \frac{D}{\sqrt{\sum_{p=1}^{T} \|g_p\|^{-2}}}.$$

**Proof**   First, observe that with $\Delta_t = -w_t \eta_t$, we have $x_{t+1} = x_t + \frac{w_{t+1:T}}{w_{1:T}}\Delta_t$. Therefore with $z_1 = x_1$ and $z_{t+1} = z_t + \Delta_t$, Theorem 1 implies:

$$\mathbb{E}[f(x_T) - f(u)] \leq \mathbb{E}\left[\frac{1}{w_{1:T}}\sum_{t=1}^{T}\langle w_t g_t, z_t - u\rangle\right].$$

Next, observe that $z_t$ is simply online gradient descent with learning rate 1 acting on the loss vectors $w_t z_t$. Standard analysis (Zinkevich, 2003) shows:

$$\sum_{t=1}^{T}\langle w_t g_t, z_t - u\rangle = \frac{\|z_1 - u\|^2}{2} - \frac{\|z_{T+1} - u\|^2}{2} + \sum_{t=1}^{T}\frac{w_t^2\|g_t\|^2}{2}.$$

This immediately implies the first part of the Theorem. Next, we need to solve for the minimizing values of $w_t$. To do this, we take the logarithm of the expression $\frac{1}{2w_{1:T}}\left(\|x_1 - u\|^2 + \sum_{t=1}^{T} w_t^2\|g_t\|^2\right)$ and differentiate:

$$\frac{\partial}{\partial w_k}\log\left[\frac{1}{2w_{1:T}}\left(\|x_1 - u\|^2 + \sum_{t=1}^{T} w_t^2\|g_t\|^2\right)\right] = \frac{2w_k\|g_k\|^2}{\|x_1 - u\|^2 + \sum_{t=1}^{T} w_t^2\|g_t\|^2} - \frac{1}{w_{1:T}}.$$

We set this equal to zero to solve for the optimal $w_k$:

$$w_k = \|g_k\|^{-2}\frac{\|x_1 - u\|^2 + \sum_{t=1}^{T} w_t^2\|g_t\|^2}{2w_{1:T}} \triangleq \lambda\|g_k\|^{-2},$$

where we have defined $\lambda = \frac{\|x_1 - u\|^2 + \sum_{t=1}^{T} w_t^2\|g_t\|^2}{2w_{1:T}}$, which does not depend on $k$. That is, the optimal $w_k$ value is proportional to $\|g_k\|^{-2}$. With this expression, we have:

$$\sum_{t=1}^{T} w_t^2\|g_t\|^2 = \lambda^2\sum_{t=1}^{T}\|g_t\|^{-2}$$

$$w_{1:T} = \lambda\sum_{t=1}^{T}\|g_t\|^{-2}.$$

So, let us now solve for $\lambda$ by plugging in these values:

$$\lambda = \frac{\|x_1 - u\|^2 + \sum_{t=1}^{T} w_t^2\|g_t\|^2}{2w_{1:T}}$$

$$\lambda = \frac{\|x_1 - u\|^2 + \lambda^2\sum_{t=1}^{T}\|g_t\|^{-2}}{2\lambda\sum_{t=1}^{T}\|g_t\|^{-2}}$$

$$\lambda = \frac{\|x_1 - u\|^2}{2\lambda\sum_{t=1}^{T}\|g_t\|^{-2}} + \frac{\lambda}{2}$$

$$\lambda = \frac{\|x_1 - u\|}{\sqrt{\sum_{t=1}^{T}\|g_t\|^{-2}}}.$$

This in turn implies the claimed optimal value for $w_k$. ∎

## E   PROOF OF THEOREM 9

**Proof**   First, observe that for any solution to the desired identity $\eta_t = \frac{w_t w_{t:1:T}}{w_{1:T}}$, replacing $w$ with $c \cdot w$ for some constant $c$ will yield a solution for $\eta$ replaced with $c \cdot \eta$. Thus, it suffices to consider the possibility that $\max_t \eta_t = 1$.

The intuition for the rest of the proof is the following: Given the value for $w_1$, we can solve for $w_{2:T}$ using the equation $\frac{w_1 w_{2:T}}{w_1 + w_{2:T}} = \eta_1$. This in turn provides us with the value of $w_{1:T}$. Now, given $w_{k:T}$ for any $k$ we can solve for $w_k$ using the equation $\eta_k = \frac{w_k w_{k+1:T}}{w_{1:T}} = \frac{w_k(w_{k:T} - w_k)}{w_{1:T}}$. Thus we may recursively compute all the values of $w$. Each of these steps requires solving a quadratic equation. We simply choose an initial $w_1$ so as to ensure that all the following quadratic equations have non-negative real roots.

Specifically, set $w_1 = \frac{2^{2T} + \sqrt{2^{4T} - 4\eta_1}}{2}$ and define $s_1 = 2^{2T}$. Then, recursively define for $t = 2, \ldots, T-1$:

$$s_t = s_{t-1} - w_{t-1}$$

$$w_t = \frac{s_t - \sqrt{s_t^2 - 4s_1\eta_t}}{2}$$

and set $w_T = s_T = s_{T-1} - w_{T-1}$. Notice that these choices satisfy:

$$w_t^2 - s_t w_t + s_1 \eta_t = 0$$

so that if we could establish (1) that all $w_t \geq 0$ and (2) $s_t = w_{t:T}$, then we would have:

$$\frac{w_t w_{t+1:T}}{w_{1:T}} = \frac{w_t(w_{t:T} - w_t)}{w_{1:T}}$$
$$= \frac{w_t(s_t - w_t)}{s_1}$$
$$= \frac{s_t w_t - w_t^2}{s_1}$$
$$= \eta_t$$

as desired.

Let us first prove (1): all $w_t \geq 0$. For $t \leq T-1$, this will hold if $s_t^2 - 4s_1\eta_t > 0$. To establish this, we will first show that $s_t \geq \frac{s_1}{2^{t-1}}$ for all $t \leq T-1$. If this holds, then we have:

$$s_t^2 \geq \frac{s_1^2}{2^{2t-2}} \geq \frac{s_1 2^{2T}}{2^{2t-2}} \geq 4s_1 \geq 4s_1\eta_t$$

for $t \leq T-1$, where we have used our assumption $\eta_t \leq 1$.

So, we now establish $s_t \geq \frac{s_1}{2^{t-1}}$ by induction for $t \leq T-1$. The statement is clear for $t = 1$. Suppose it holds for for all $t \leq k$ for some $k \leq T-2$. Then we have:

$$s_{k+1} = s_k - w_k$$
$$= \frac{s_k + \sqrt{s_k^2 - 4s_1\eta_k}}{2}$$
$$\geq \frac{s_k}{2}$$
$$\geq \frac{s_1}{2^{k-1+1}},$$

which establishes the claim. Therefore for $t \leq T-1$, $w_t$ are non-negative real numbers. Finally, we have:

$$w_{T-1} = \frac{s_{T-1} + \sqrt{s_{T-1}^2 - 4s_1\eta_t}}{2} \leq s_{T-1},$$

so that $w_T = s_T = s_{T-1} - w_{T-1} \geq 0$. Thus $w_t \geq 0$ for all $t$.

Next, to show (2): $s_t = w_{t:T}$. This is nearly immediate. By definition we have $w_T = s_T$. Suppose $s_k = w_{t:T}$ for some $t \geq 2$. Then $s_t = s_{t-1} - w_{t-1}$ so that $s_{t-1} = s_t + w_{t-1} = w_{t-1:T}$. Thus by induction we have $s_t = w_{t:T}$ for all $t$, which is the last thing we needed to show. ∎

## F    SCHEDULES FOR PER-COORDINATE UPDATES

Many popular optimization algorithms in use today like Adam (Kingma & Ba, 2015) and its variants employ *per-coordinate* updates: the update $\Delta_t$ is not proportional to the gradient $g_t$ but instead scales each coordinate of $g_t$ by an adaptively chosen value. In this section we propose an approximately optimal schedule for such methods.

**Theorem 8** *Suppose $\Delta_t = z_{t+1} - z_t = -w_t \cdot (\eta_t \odot g_t)$ where $\eta_t \in \mathbb{R}^d$ is a vector of learning rates and $\odot$ indicates coordinate-wise product. Set $x_{t+1} = x_t - \frac{w_{t+1:T}}{w_{1:T}} \Delta_t$. Define the quantity $R$ by:*

$$R = \sqrt{\sum_{t=1}^{T} \sum_{i=1}^{d} \frac{(z_{t,i} - u_i)^2 - (z_{t+1,i} - u_i)^2}{\eta_{t,i}}}$$

*Then we have:*

$$\mathbb{E}[f(x_T) - f(u)] \leq \mathbb{E}\left[\frac{1}{w_{1:T}} \left(\frac{R^2}{2} + \sum_{t=1}^{T} w_t^2 \sum_{i=1}^{d} \eta_{t,i} g_{t,i}^2\right)\right].$$

*Moreover, for any given fixed values for $R$ and $g_{t,i}^2$, the setting of $w_t$ that minimizes the expression $\frac{1}{w_{1:T}} \left(\frac{R^2}{2} + \sum_{t=1}^{T} w_t^2 \sum_{i=1}^{d} \eta_{t,i}^2 g_{t,i}^2\right)$ is:*

$$w_t = \frac{R}{\sqrt{\sum_{t=1}^{T} \left(\sum_{i=1}^{d} \eta_{t,i} g_{t,i}^2\right)^{-1}}} \cdot \left(\sum_{i=1}^{d} \eta_{t,i} g_{t,i}^2\right)^{-1}.$$

As an example of how to use this result, let us suppose we are employing the Adam optimizer, and let us also also ignore the presence of momentum when computing the weights (so really, these will be the weights for the RMSProp optimizer). In this case, $\eta_{t,i} \propto \frac{1}{\sqrt{v_{t,i}}}$ where $v_{t,i}$ is the exponential average of the squared gradients. Thus, we get:

$$w_t = \lambda \left(\sum_{i=1}^{d} \frac{g_{t,i}^2}{\sqrt{v_{t,i}}}\right)^{-1}$$

for some $\lambda$. Then, we use a learning rate schedule of $\frac{w_t w_{t+1:T}}{w_{1:T}}$.

That is, the corresponding procedure to Algorithm 1 is given by Algorithm 2:

In practice, however, recording the value $w_t \propto \left(\sum_{i=}^{d} \frac{g_{t,i}^2}{\sqrt{v_{t,i}}}\right)^{-1}$ may be difficult (for example, in Pytorch, the Adam implementation is primarily C code rather than Python, which makes it substantially more involved to modify). However, by inspecting the formula for $w_t$, we can see that it is likely to be an interpolation between the $1/\|g_t\|_2^2$ and $1/\|g_t\|_1$ (the L1 norm is not squared). The intuition behind this is that if $v_{t,i}$ has a minimum value of $|g_{t,i}|$. With this minimum value, $w_t \propto 1/\|g_t\|_1$. On the other hand if all $v_{t,i}$ are the same constant, then $w_t \propto 1/\|g_t\|_2^2$. In practice we expect behavior closer to the first case and recommend using $w_t \propto 1/\|g_t\|_1$.

---

**Algorithm 2** Schedule Refinement for Adam

---

1: **Input:** $G = \left( G_t = \sum_{i=1}^{d} \frac{g_{t,i}^2}{\sqrt{v_{t,i}}} \right)$ length $T$ sequence of weighted gradient norms from Adam optimizer, smoothing hyper-parameter $\tau > 0$
2: $\hat{G} = \text{median\_filter}(G, \text{filter\_width} = \tau T, \text{padding} = (\text{nearest}, \text{reflect}))$
3: Define $w_t = \hat{G}_t^{-1}$
4: For each $t$, let:
5:

$$\eta_t = w_t \sum_{p=t+1}^{T} w_p$$

6: Return normalized schedule $\eta / \max(\eta)$

---

**Proof** [proof of Theorem 8] By standard online gradient descent analysis, we have:

$$\langle w_t \cdot g_t, z_t - u \rangle = \sum_{i=1}^{d} w_t g_{t,i}(z_{t,i} - u_i)$$

$$= \sum_{i=1}^{d} \frac{(z_{t,i} - u_i)^2}{2\eta_{t,i}} - \frac{(z_{t+1,i} - u_i)^2}{2\eta_{t,i}} + \frac{\eta_{t,i}}{2} w_t^2 g_{t,i}^2.$$

Summing this over $t$ from $1$ to $T$, we get

$$\sum_{t=1}^{T} \langle w_t \cdot g_t, z_t - u \rangle = \sum_{t=1}^{T} \sum_{i=1}^{d} \frac{(z_{t,i} - u_i)^2}{2\eta_{t,i}} - \frac{(z_{t+1,i} - u_i)^2}{2\eta_{t,i}} + \sum_{t=1}^{T} w_t^2 \sum_{i=1}^{d} \frac{\eta_{t,i} g_{t,i}^2}{2}$$

$$= \frac{R^2}{2} + \sum_{t=1}^{T} \frac{w_t^2}{2} \sum_{i=1}^{d} \eta_{t,i} g_{t,i}^2.$$

The first part of the result now follows from Theorem 1.

Next, we again take the logarithm, differentiate and set equal to zero:

$$0 = \frac{\partial}{\partial w_k} \log \left( \frac{1}{w_{1:T}} \left( \frac{R^2}{2} + \sum_{t=1}^{T} \frac{w_t^2}{2} \sum_{i=1}^{d} \eta_{t,i} g_{t,i}^2 \right) \right)$$

$$= \frac{2 w_k \sum_{i=1}^{d} \eta_{k,i}^2 g_{k,i}^2}{R^2 + \sum_{t=1}^{T} w_t^2 \sum_{i=1}^{d} \eta_{t,i} g_{t,i}^2} - \frac{1}{w_{1:T}}.$$

Rearranging, we obtain

$$w_k = \frac{\left( \sum_{i=1}^{d} \eta_{k,i}^2 g_{k,i}^2 \right)^{-1} \left( R^2 + \sum_{t=1}^{T} w_t^2 \sum_{i=1}^{d} \eta_{t,i} g_{t,i}^2 \right)}{2 w_{1:T}}$$

$$\triangleq \lambda \left( \sum_{i=1}^{d} \eta_{k,i} g_{k,i}^2 \right)^{-1},$$

where we have collected the non $k$-dependent terms into $\lambda = \frac{R^2 + \sum_{t=1}^{T} w_t^2 \sum_{i=1}^{d} \eta_{t,i} g_{t,i}^2}{2w_{1:T}}$. Now we solve for $\lambda$:

$$
\begin{aligned}
\lambda &= \frac{R^2 + \sum_{t=1}^{T} w_t^2 \sum_{i=1}^{d} \eta_{t,i} g_{t,i}^2}{2w_{1:T}} \\
&= \frac{R^2 + \lambda^2 \sum_{t=1}^{T} \left( \sum_{i=1}^{d} \eta_{t,i} g_{t,i}^2 \right)^{-1}}{2\lambda \sum_{t=1}^{T} \left( \sum_{i=1}^{d} \eta_{t,i} g_{t,i}^2 \right)^{-1}} \\
&= \frac{R^2}{2\lambda \sum_{t=1}^{T} \left( \sum_{i=1}^{d} \eta_{t,i} g_{t,i}^2 \right)^{-1}} + \frac{\lambda}{2} \\
&= \frac{R}{\sqrt{\sum_{t=1}^{T} \left( \sum_{i=1}^{d} \eta_{t,i} g_{t,i}^2 \right)^{-1}}}.
\end{aligned}
$$

Putting $w_t = \lambda \left( \sum_{i=1}^{d} \eta_{t,i} g_{t,i}^2 \right)^{-1}$ completes the proof. ∎

## G  EXPLICIT BOUNDS FOR ARBITRARY SCHEDULES

Section 2 develops a framework that suggests using learning rates of the form $\eta_t = \frac{w_t w_{t+1:T}}{w_{1:T}}$ and in Theorem 3 we provided a simple closed-form solution for the optimal values of $w_t$. Given this apparently restricted form of $\eta_t$, it is natural to ask if this restriction is real: that is, can *every* schedule $\eta_1, \ldots, \eta_{T-1}$ be represented using some weights $w_t$? Theorem 9 shows that the answer is "yes":

**Theorem 9** *Let $\eta_1, \ldots, \eta_{T-1}$ be a sequence of non-negative numbers with $\eta_1 \geq 0$. Then there is a sequence of non-negative weights $w_1, \ldots, w_T$ such that $\eta_t = \frac{w_t w_{t+1:T}}{w_{1:T}}$ for all $t$.*

This Theorem shows that by optimizing the weights $w_t$, we are in some sense also solving for the optimal learning rate $\eta_t$. However, notice that the reverse is not true: any given schedule $\eta_t$ can be represented by a number of different weights $w_t$, and these weights give rise to different bounds using Theorem 1. The proof of Theorem 9 works by constructing a particular set of weights $w_t$, but these may not be the best weights in terms of providing the tightest convergence bound for the given $\eta_t$.

Although Theorem 9 shows that the learning rate representation of Theorem 3 indeed covers all possible schedules, it does not provide a user-friendly way to analyze the convergence of an arbitrary schedule. In this section, we provide an alternative analysis that fills this gap.

This approach is closer to previous final-iterate analysis techniques: we bound the final iterate in terms of the average iterate, plus an additional *error* term. This bound is strictly looser than those of Theorems 1 and 3 in the constant factors, but provides a convenient way to analyze arbitrary schedules. The bound is presented in Theorem 10 below.

**Theorem 10** *Suppose that $f$ is convex and let $x_t$ be given by SGD with learning rates $\eta_t$: $x_{t+1} = x_t - \eta_t g_t$. Then for the last iterate $x_T$ we have:*

$$
\begin{aligned}
\mathbb{E}[f(x_T) - f(u)] \leq \mathbb{E} &\left[ \frac{1}{2\sum_{t=1}^{T} \eta_t} D^2 + \frac{1}{2\sum_{t=1}^{T} \eta_t} \sum_{t=1}^{T} \eta_t^2 \|g_t\|^2 \right] \\
&+ \mathbb{E} \left[ \frac{1}{2} \sum_{k=1}^{T-1} \frac{\eta_k}{\sum_{t=k+1}^{T} \eta_t} \left( \frac{1}{\sum_{t=k}^{T} \eta_t} \sum_{t=k}^{T} \eta_t^2 \|g_t\|^2 \right) \right].
\end{aligned}
\tag{4}
$$

Optimizing this bound with respect to the step-size sequence produces schedules that are visually indistinguishable from those of the regret based approach described in Section 2.1 for large $T$.

To prove Theorem 10, we will need to control quantities like:

$$\sum_{t=k}^{T} \eta_t \left(q_t - q_k\right) = f(x_t) - f(x_k).$$

This can be bounded by the usual suboptimality inequality for SGD/GD methods, which holds for any $u$:

$$\sum_{t=k}^{T} \eta_t \left[f(x_t) - f(u)\right] \le \frac{1}{2} \|x_k - u\|^2 + \frac{1}{2} \sum_{t=k}^{T} \eta_t^2 \|g_t\|^2,$$

Setting $u = x_k$ yields:

$$\sum_{t=k}^{T} \eta_t \left[q_t - q_k\right] \le \frac{1}{2} \sum_{t=k}^{T} \eta_t^2 \|g_t\|^2.$$

We can use this in our Lemma 5 to obtain the following result:

**Corollary 11**

$$q_T = \frac{1}{\sum_{t=1}^{T} \eta_t} \sum_{t=1}^{T} \eta_t q_t + \frac{1}{2} \sum_{k=1}^{T-1} \frac{\eta_k}{\sum_{t=k+1}^{T} \eta_t} \left(\frac{1}{\sum_{t=k}^{T} \eta_t} \sum_{t=k}^{T} \eta_t^2 \|g_t\|^2\right).$$

Now, we are ready to prove Theorem 10.

**Theorem 10** *Suppose that $f$ is convex and let $x_t$ be given by SGD with learning rates $\eta_t$: $x_{t+1} = x_t - \eta_t g_t$. Then for the last iterate $x_T$ we have:*

$$\mathbb{E}[f(x_T) - f(u)] \le \mathbb{E}\left[\frac{1}{2\sum_{t=1}^{T} \eta_t} D^2 + \frac{1}{2\sum_{t=1}^{T} \eta_t} \sum_{t=1}^{T} \eta_t^2 \|g_t\|^2\right]$$

$$+ \mathbb{E}\left[\frac{1}{2} \sum_{k=1}^{T-1} \frac{\eta_k}{\sum_{t=k+1}^{T} \eta_t} \left(\frac{1}{\sum_{t=k}^{T} \eta_t} \sum_{t=k}^{T} \eta_t^2 \|g_t\|^2\right)\right]. \tag{4}$$

**Proof** Following the standard convergence bound approach:

$$\begin{aligned}
\|x_{t+1} - u\|^2 &= \|x_t - \eta_t g_t - u\|^2 \\
&= \|x_t - u\|^2 - 2\eta_t \langle g_t, x_t - u \rangle + \eta_t^2 \|g_t\|^2 \\
&\le \|x_t - u\|^2 - 2\eta_t \left[f(x_t) - f(u)\right] + \eta_t^2 \|g_t\|^2.
\end{aligned}$$

Summing over $t$ and telescoping gives:

$$\sum_{t=1}^{T} \eta_t \left[f(x_t) - f(u)\right] \le \frac{1}{2} D^2 + \sum_{t=1}^{T} \eta_t^2 \|g_t\|^2.$$

Then divide through by $\sum_{t=1}^{T} \eta_t$:

$$\frac{1}{\sum_{t=1}^{T} \eta_t} \sum_{t=1}^{T} \eta_t \left[f(x_t) - f_*\right] \le \frac{1}{2\sum_{t=1}^{T} \eta_t} D^2 + \frac{1}{2\sum_{t=1}^{T} \eta_t} \sum_{t=1}^{T} \eta_t^2 \|g_t\|^2.$$

Now we apply Corollary 11 to get:

$$f(x_T) - f_* \le \frac{1}{2\sum_{t=1}^{T} \eta_t} D^2 + \frac{1}{2\sum_{t=1}^{T} \eta_t} \sum_{t=1}^{T} \eta_t^2 \|g_t\|^2$$

$$+ \frac{1}{2} \sum_{k=1}^{T-1} \frac{\eta_k}{\sum_{t=k+1}^{T} \eta_t} \left(\frac{1}{\sum_{t=k}^{T} \eta_t} \sum_{t=k}^{T} \eta_t^2 \|g_t\|^2\right). \qquad \blacksquare$$

By using a worst-case bound of $\|g_t\|^2 \le G^2$, we obtain:

**Corollary 12** *If $f$ is $G$-Lipschitz, then*

$$f(x_T) - f_* \leq \frac{1}{2\sum_{t=1}^{T} \eta_t} D^2 + \frac{G^2}{2\sum_{t=1}^{T} \eta_t} \sum_{t=1}^{T} \eta_t^2$$
$$+ \frac{G^2}{2} \sum_{k=1}^{T-1} \frac{\eta_k}{\sum_{t=k+1}^{T} \eta_t} \left( \frac{1}{\sum_{t=k}^{T} \eta_t} \sum_{t=k}^{T} \eta_t^2 \right).$$

## G.1 LINEAR SCHEDULE ANALYSIS WITH THEOREM 10

In this section, we use Theorem 10 to re-analyze the the linear decay schedule:

$$\eta_t = \frac{D}{G\sqrt{T}} \left( 1 - \frac{t}{T+1} \right). \tag{5}$$

The resulting convergence rate is asymptotically correct, but does not achieve the optimal constant obtained by Theorem 3.

**Theorem 13** *Schedule 5 gives the following bound on the last iterate:*

$$f(x_T) - f_* \leq \left( 2 + \frac{1}{4} \right) \frac{DG}{\sqrt{T}},$$

*or more precisely:*

$$f(x_T) - f_* \leq \left( 2 + \frac{H(T-1) - 2/3}{T+1} \right) \frac{DG}{\sqrt{T}},$$

*where $H(T)$ is the Tth harmonic sum.*

**Proof** We start with the above bound:

$$f(x_T) - f_* \leq \frac{1}{2\sum_{t=1}^{T} \eta_t} D^2 + \frac{G^2}{2\sum_{t=1}^{T} \eta_t} \sum_{t=1}^{T} \eta_t^2$$
$$+ \frac{G^2}{2} \sum_{k=1}^{T-1} \frac{\eta_k}{\sum_{t=k+1}^{T} \eta_t} \left( \frac{1}{\sum_{t=k}^{T} \eta_t} \sum_{t=k}^{T} \eta_t^2 \right).$$

Since the $\frac{D}{G\sqrt{T}}$ part of the step size is constant, it can be pulled outside the summations, so we just need to focus on summations involving $\left( 1 - \frac{t}{T+1} \right)$. For the first two terms we have in the bound, they simplify

$$\sum_{t=1}^{T} \eta_t \propto \sum_{t=1}^{T} \left( 1 - \frac{t}{T+1} \right)$$
$$= \left( T - \frac{1}{T+1} \sum_{t=1}^{T} t \right)$$
$$= \left( T - \frac{1}{2} \frac{T(T+1)}{T+1} \right)$$
$$= \frac{T}{2}.$$

Similarly,

$$
\begin{aligned}
\sum_{t=1}^{T} \eta_t^2 &\propto \sum_{t=1}^{T} \left(1 - \frac{t}{T+1}\right)^2 \\
&= \sum_{t=1}^{T} \left(1 - \frac{2t}{T+1} + \frac{t^2}{T+1^2}\right) \\
&= \left(T - T + \sum_{t=1}^{T} \frac{t^2}{T^2}\right) \\
&= \left(\frac{T(T+1)(2T+1)}{6(T+1)^2}\right) \\
&= \left(\frac{T(T+1)(T+1/2)}{3(T+1)^2}\right) \\
&\leq \frac{T}{3}.
\end{aligned}
$$

So we have the following bound on the last iterate:

$$
\begin{aligned}
f(x_T) - f_* \leq \frac{DG}{\sqrt{T}} + \frac{DG}{3\sqrt{T}} \\
+ \frac{G^2}{2} \sum_{k=1}^{T-1} \frac{\eta_k}{\sum_{t=k+1}^{T} \eta_t} \left(\frac{1}{\sum_{t=k}^{T} \eta_t} \sum_{t=k+1}^{T} \eta_t^2\right).
\end{aligned}
$$

To simplify the remaining term we rely on computer algebra software. SymPy gives:

$$
\sum_{k=1}^{T-1} \frac{\eta_k}{\sum_{t=k+1}^{T} \eta_t} \left(\frac{1}{\sum_{t=k}^{T} \eta_t} \sum_{t=k+1}^{T} \eta_t^2\right) = \left(\frac{D}{G\sqrt{T}}\right) \frac{4T + 6H\left(T-1\right) - 4}{3(T+1)}.
$$

Where $H(T) = \sum_{t=1}^{T} 1/t$ is the harmonic sum. So:

$$
\begin{aligned}
f(x_T) - f_* &\leq \frac{DG}{\sqrt{T}} + \frac{DG}{3\sqrt{T}} + \frac{2DG\sqrt{T}}{3T} + \frac{DG\left(3H\left(T-1\right) - 2\right)}{3\left(T+1\right)\sqrt{T}} \\
&\leq \frac{2DG}{\sqrt{T}} + \frac{DG}{3\sqrt{T}} + \frac{DG\left(3H\left(T-1\right) - 2\right)}{3(T+1)\sqrt{T}}.
\end{aligned}
$$

Note that the term $\left(3H\left(T-1\right) - 2\right)/3\left(T+1\right) \leq 1/4$ for all $T$, so:

$$
\frac{DG\left(3H\left(T-1\right) - 2\right)}{3(T+1)\sqrt{T}} = \frac{DG}{4\sqrt{T}},
$$

So combining with the $\frac{DG}{\sqrt{T}} + \frac{DG}{3\sqrt{T}}$ terms, we have:

$$
f(x_T) - f_* \leq \left(2 + \frac{1}{4}\right) \frac{DG}{\sqrt{T}}.
$$

bounding the harmonic function with a log gives instead:

$$
f(x_T) - f_* \leq \frac{2DG}{\sqrt{T}} + O\left(\frac{DG\log(T)}{T^{3/2}}\right).
$$

$\blacksquare$

# H EXPERIMENTAL SETUP

Our experiments on CIFAR-10, CIFAR-100, ImageNet and RCNN use SGD, and the remaining problems use Adam. We used decoupled weight decay with Adam in each case following standard practice for each problem.

## H.1 CONVEX EXPERIMENTS

Each dataset is obtained from the LIBSVM repository with used without modifications.

| Hyper-parameter | Value |
|---|---|
| GPUs | 1×V100 |
| Batch size | 16 |
| Epochs | 100 |
| Seeds | 10 |

| Hyper-parameter | Value |
|---|---|
| Decay | 0.0 |
| Optimizer | Adam |
| $\beta_1$ | 0.9 |
| $\beta_2$ | 0.95 |

## H.2 CIFAR-10

We used custom training code based on the PyTorch tutorial code for this problem. Following standard data-augmentation practises, we applied random horizontal flips and random offset cropping down to 32x32, using reflection padding of 4 pixels. Input pixel dataD was normalized by centering around 0.5.

| Hyper-parameter | Value |
|---|---|
| Architecture | Wide ResNet 16-8 |
| Epochs | 300 |
| GPUs | 1×V100 |
| Batch size per GPU | 128 |

| Hyper-parameter | Value |
|---|---|
| Seeds | 10 |
| decay | 0.0001 |
| Momentum | 0.9 |

## H.3 CIFAR-100

We used the same codebase as for our CIFAR-10 experiments, with the same data augmentation.

We normalized each input image using fixed mean and standard error values derived from preprocessing the data.

| Hyper-parameter | Value |
|---|---|
| Architecture | DenseNet [6,12,24,16], growth rate 12 |
| Epochs | 300 |
| GPUs | 1×V100 |

| Hyper-parameter | Value |
|---|---|
| Batch size per GPU | 64 |
| Seeds | 10 |
| Decay | 0.0002 |
| Momentum | 0.9 |

## H.4 IMAGENET

We used the same code-base as for our CIFAR-10 experiments, and applied the same preprocessing procedure. The data-augmentations consisted of PyTorch's RandomResizedCrop, cropping to 224x224 followed by random horizontal flips. Test images used a fixed resize to 256x256 followed by a center crop to 224x224.

| Hyper-parameter | Value |
|---|---|
| Architecture | ResNet50 |
| Epochs | 100 |
| GPUs | 8×V100 |
| Batch size per GPU | 32 |

| Hyper-parameter | Value |
|---|---|
| Seeds | 5 |
| Decay | 0.0001 |
| Momentum | 0.9 |

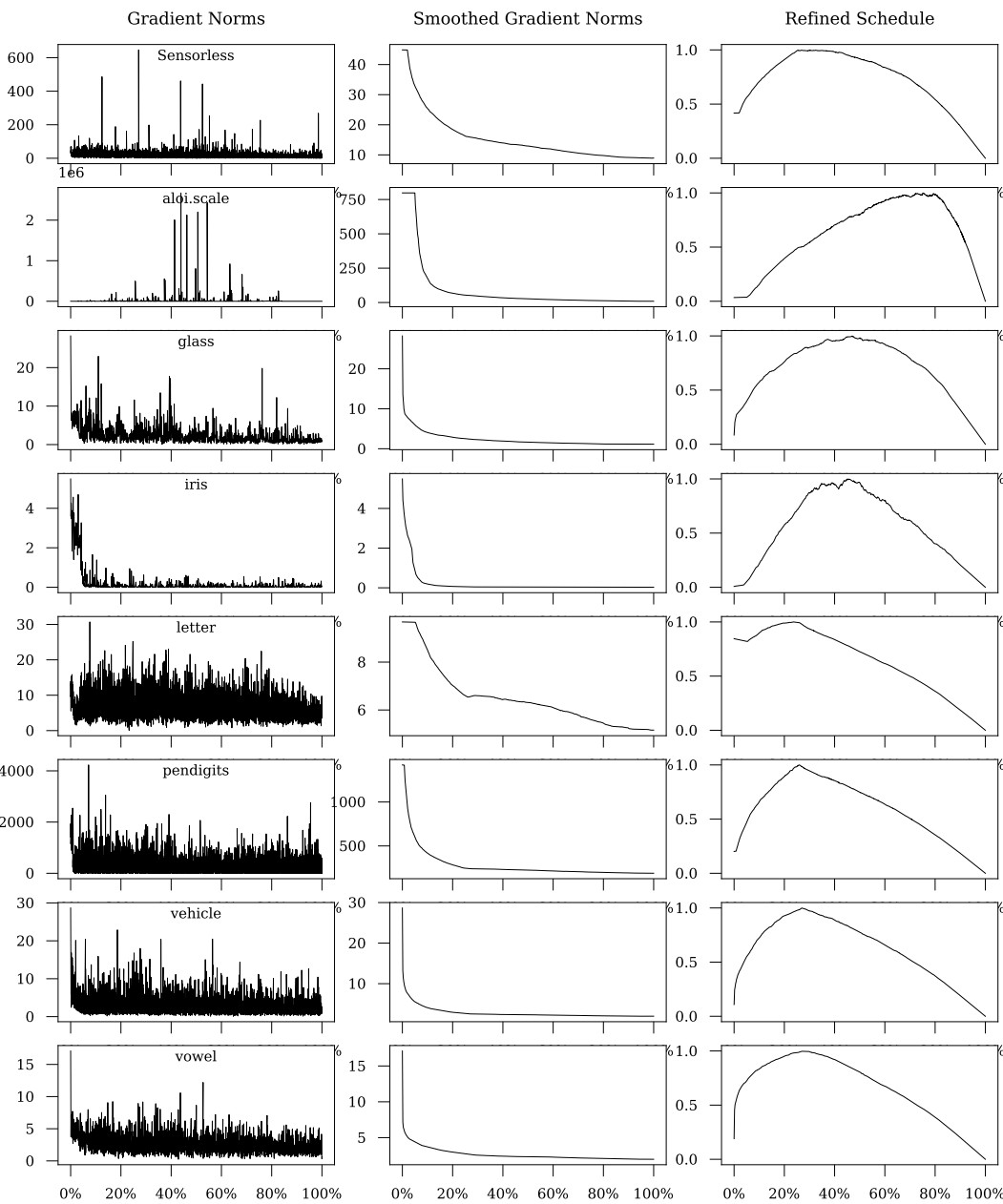

Figure 6: Logistic regression schedules, generated using a linear decay schedule with warmup for the initial run.

## H.5   IWSLT14

We used the FairSeq framework [1] for our experiments here, as well as for our GPT and RoBERTa experiments. Rather than a vanilla LSTM we use the variant from (Wiseman & Rush, 2016) provided in the FairSeq codebase.

| Hyper-parameter | Value |
| --- | --- |
| Architecture | lstm_wiseman_iwslt_de_en |
| Max Epoch | 55 |
| GPUs | 1×V100 |
| Tokens per batch | 4096 |
| Warmup steps | 4000 |
| Dropout | 0.3 |
| Label smoothing | 0.1 |

| Hyper-parameter | Value |
| --- | --- |
| Share decoder, input, output embed | True |
| Float16 | True |
| Update Frequency | 1 |
| Seeds | 10 |
| Decay | 0.05 |
| $\beta_1, \beta_2$ | 0.9, 0.98 |

## H.6   ROBERTA

The RoBERTa implementation in FairSeq is the canonical one. We differ from the paper's results by training for a shorter duration, which is necessary to keep our experiments computationally tractable. Our BookWiki dataset matches the original paper.

| Hyper-parameter | Value |
| --- | --- |
| Architecture | roberta_base |
| Task | masked_lm |
| Max updates | 23,000 |
| GPUs | 8×V100 |
| Max tokens per sample | 512 |
| Dropout | 0.1 |
| Attention Dropout | 0.1 |
| Max sentences | 16 |

| Hyper-parameter | Value |
| --- | --- |
| Warmup | 10,000 |
| Sample Break Mode | Complete |
| Float16 | True |
| Update Frequency | 16 |
| Seeds | 5 |
| Decay | 0.0 |
| $\beta_1, \beta_2$ | 0.9, 0.98 |

## H.7   GPT

Since the training dataset for GPT models are not availiable, we use the BookWiki dataset as used for RoBERTa training. Our model here is small, using 12 decoding layers and a decoder embedding dim of 768, giving 162 million parameters.

| Hyper-parameter | Value |
| --- | --- |
| Architecture | transformer_lm_gpt |
| Task | language_modeling |
| Max updates | 65,000 |
| GPUs | 8×V100 |
| Tokens per sample | 512 |
| Dropout | 0.1 |
| Attention Dropout | 0.1 |
| Max sentences | 1 |
| Warmup | 10,000 |

| Hyper-parameter | Value |
| --- | --- |
| Sample Break Mode | Complete |
| Share decoder, input, output embed | True |
| Float16 | True |
| Update Frequency | 16 |
| Seeds | 5 |
| Decay | 0.005 |
| $\beta_1, \beta_2$ | 0.9, 0.98 |

## H.8   VIT

Our implementation uses the PyTorch Image Models library [2], with hyper-parameters following examples given in the repository.

---

[1] https://github.com/facebookresearch/fairseq
[2] https://github.com/rwightman/pytorch-image-models

| Hyper-parameter | Value |
|---|---|
| Model | vit_tiny_patch16_224 |
| Epochs | 300 |
| Batch Size | 512 |
| Warmup epochs | 5 |
| Hflip | 0.5 |
| aa | rand-m6-mstd0.5 |
| mixup | 0.1 |

| Hyper-parameter | Value |
|---|---|
| mixup | 0.1 |
| cutmix | 1.0 |
| Crop Pct | 0.9 |
| BCE Loss | True |
| Seeds | 5 |
| Decay | 0.1 |
| $\beta_1, \beta_2$ | 0.9, 0.999 |

## H.9 DLRM

We used a custom implementation of the DLRM model based on the publicly available code. Our optimizer uses dense gradients for implementation simplicity, although sparse-gradients using Ada-Grad is a more common baseline on this problem, we consider AdaGrad variants of our scheduling approach as future work.

| Hyper-parameter | Value |
|---|---|
| Iterations | 300 000 |
| Batch Size | 128 |
| Emb Dimension | 16 |

| | |
|---|---|
| Seeds | 5 |
| Decay | 0.0 |
| $\beta_1, \beta_2$ | 0.9, 0.999 |

## H.10 MRI

We used the version of the the fastMRI code base at `https://github.com/facebookresearch/fastMRI/tree/main/banding_removal`. Note that we found that training failed using PyTorch 2 or newer, and so we ran these experiments using PyTorch 1.9.

| Hyper-parameter | Value |
|---|---|
| Architecture | 12 layer VarNet 2.0 |
| Epochs | 50 |
| GPUs | 8×V100 |
| Batch size per GPU | 1 |
| Acceleration factor | 4 |

| | |
|---|---|
| Low frequency lines | 16 |
| Mask type | Offset-1 |
| Seeds | 5 |
| Decay | 0.0 |
| $\beta_1, \beta_2$ | 0.9, 0.999 |

## H.11 RCNN

Our RCNN experiments use Detectron2[3], and we use pretrained ResNet backbone[4].

| Hyper-parameter | Value |
|---|---|
| Backbone | ResNet-50 |
| Max Iter | 200000 |
| IMS Per Batch | 16 |

| Hyper-parameter | Value |
|---|---|
| Momentum | 0.9 |
| Decay | 1.5e-4 |

---

[3] `https://github.com/facebookresearch/detectron2`
[4] `detectron2://ImageNetPretrained/MSRA/R-50.pkl`

Table 6: Popularity of standard learning rate schedules.

| PyTorch Scheduler | Github Files (in thousands) |
|---|---|
| ReduceLROnPlateau | 105 |
| StepLR | 101 |
| MultiStepLR | 37.9 |
| CosineAnnealingLR | 37.1 |
| ExponentialLR | 16 |
| OneCycleLR | 14.9 |
| CosineAnnealingWarmRestarts | 10.9 |
| CyclicLR | 9.1 |
| LinearLR | 5.9 |
| ConstantLR | 3.6 |
| MultiplicativeLR | 2.6 |
| PolynomialLR | 1.3 |

