# OpenReview forum: "When, Why and How Much? Adaptive Learning Rate Scheduling by Refinement"
_ICLR.cc/2024/Conference — Submitted to ICLR 2024_

### Official Review · Reviewer_CaT7 · 2023-10-30

**Soundness:** 2 fair
**Presentation:** 3 good
**Contribution:** 2 fair
**Rating:** 5
**Confidence:** 3

**Summary:**

This paper proposes a refined study of learning rate schedule for SGD. It presents last iterate convergence results. The proposed method automatically yields learning rate warm-up and rapid learning rate annealing near the end of training. The authors also conduct comprehensive numerical experiment to illustrate the performance of the proposed method.

**Strengths:**

This work proposes a novel refinement method, which uses a prior training run to produce an improved schedule to use in future runs. This method is guaranteed in a last iterate convergence fashion and can be generalized beyond SGD, which are more realistic.
Based on this method, a practical method is proposed. Comprehensive experiments validate the performance of the schedule refinement method.

**Weaknesses:**

The theory appears to be more of a heuristic that doesn't directly address practical implementation.
Some places need further clarifications.

**Questions:**

1.  In Figure 1, how do you define warm-up? Why the refined schedule starts from nearly zero? Does this lead to slow update at the beginning? Is there any way this can be improved?
2. In the analysis of Th 3, $w_t$ and $g_t$ are assumed to be conditionally independent. Based on that, equation (2) presents a last iteration problem-dependent regret bound. However, the chosen $w_t$ does not satisfy the independent assumption, which means equation (2) does not hold. How you argue this setting of $w_t$ still minimizes the bound?  What is the impact here?
3. What is the definition of median_filter? How it ensures the gradient norm sequence does not change significantly after refinement?
4. It seems that the theory is weak, only useful in a heuristic way. What is the novelty in your proof?
 In addition, it is restricted to convex functions. How about cases of non-convex functions under further assumptions?

---

> ### Author Response · Authors · 2023-11-13
> **Response**
>
> Thanks for these questions, they really hit at the heart of our paper. We will try and address them individually.
> * Our schedules actually start slightly above 0, with the precise value depending on the norm sequence, although the difference from 0 is not perceptible in the plots unless you zoom in a lot. The use of such small initial LR here is similar to those used in major frameworks such as FairSeq that we use for our GPT and BERT experiments, and as far as we are aware is the most common approach.
> * This is the major subtlety in our work, thank you for asking for clarification on this. Due to the dependence, this setting is only exactly usable with hindsight. However, by using the gradient norms from a previous run, rather than the current run, there is no longer a dependence. As long as the rough shape of the gradient norm sequence matches on the second run, then our bound gives us an improvement. We use a smoothing filter to avoid a dependence on the exact gradient norm, only the smoothed shape of the sequence.
> We can't guarantee that the gradient norm sequence stays similar, but this appears to hold strongly in practice. The key point is that it only needs to hold approximately for the bound to be an improvement, and we can even calculate the improvement after the run! We find that our bound is between 9 to 1.5 times better than the bound for the linear decay schedule across the set of deep learning problems we considered.
> * Our median filter is precisely the implementation in SciPy. Thank you for pointing this out, we forgot to include the precise definition!
> * Our theory has a major theoretical novelty: it is the first to establish that for any weight sequence for the regret, there is a corresponding *closed form* learning rate schedule that achieves last iterate convergence matching with the optimal rate possible for that weight sequence. This was not known before, and indeed many people we have discussed this result with thought that this was impossible!
> * The surprising fact that the linear decay schedule is *exactly* optimal for non-smooth convex optimization (even without weighting) could also be considered a novelty of our work, although it also appears in concurrent work by Zamani and Glineur published online in late July (which is within the concurrent work window for ICLR).
>
> * While our theory is restricted to the convex setting, it is already known that last-iterate rates for the general stochastic non-convex setting are impossible (https://arxiv.org/pdf/1910.01845.pdf Sec 3.1). It's remarkable that the convex analysis yields methods that work both across the convex and nonconvex settings, and we don't yet understand why. We do see a larger advantage in our convex experiments from using refinement methods and in our non-convex experiments.
>
> Please let us know if you have any further questions, we see that you have a generally favorable review of our work, and we hope you will consider our work as publication worthy.

---

### Official Review · Reviewer_HeFS · 2023-11-01

**Soundness:** 3 good
**Presentation:** 3 good
**Contribution:** 3 good
**Rating:** 6
**Confidence:** 2

**Summary:**

The paper proposes a set of learning rate schedules in deep learning from convex optimization theory. The theory can partially explain the popular linear decay schedule and highlights the importance to decrease the learning rate at the ending phase of training. The paper also evaluates the performance of different learning rate schedules on an extensive set of experiments and supports the use of linear-decay learning rate schedules.

**Strengths:**

The paper proposes an interesting theoretical framework to explain the linear decay schedule in deep learning. The theory can also be used to build the learning rate schedule for not only SGD, but also element-wise algorithms like Adam. Numerically, the paper introduces a series of new techniques in algorithm 1 and 2, including median filter, $\ell_2$ norm inverse weighting, and $\ell_1$ norm inverse weighting.

**Weaknesses:**

1. The authors should emphasize that their theory is based on convex optimization. In theorem 3 and 8, the assumption that $f$ is a convex function should be clearly stated.

2. For experiments in section 3.2, it is not very clear which algorithm is used. Is it SGD with $\ell_2$ norm inverse weighting or Adam with $\ell_1$ norm inverse weighting?

3. Also, authors can provide more details about "a sweep of learning rates on a grid" for the refined schedule. Why should we sweep over any parameter in the refined schedule? Based on algorithm 1 or 2, the refined schedule does not contain any additional tuning parameter than $\tau$.

**Questions:**

It seems that the schedule refinement requires two times computation resources than the standard linear decay or cosine schedule. Is there a way to improve that?

---

> ### Author Response · Authors · 2023-11-13
> **Response**
>
> Thank you for the detailed comments. As you say, we are missing some details regarding our theoretical assumptions and the specific setup of the experiments. We address these points individually below. We will also update the camera ready with these answers.
>
> * You're absolutely correct that we only analyze learning rate schedules in the convex setting. We will update the statement of both Theorems to reflect this clearly. Learning rates overall are not as well understood in the non-convex setting, and we are actively working on extending our work to that setting, but the theoretical obstacles are formidable. Currently last-iterate bounds are known to be impossible for the general stochastic non-convex setting (https://arxiv.org/pdf/1910.01845.pdf Sec 3.1), so a different approach is required.
> * For the experiments in Section 3.2, we use inverse/L2 norm squared for the SGD problems (ImageNet, RCNN, CIFAR) and Adam for the remaining problems, with inverse L1 norm. Currently this is not clearly stated in the experiments section but partially discussed in Section 2.1. We apologize.
> * For the learning rate sweep, we used a [1, 2, 5]]x10^i grid, for varying i. We sweep the baseline learning rate, the quantity that multiplies the schedule, for every problem. We include the sweep here for the refinement schedule also, so the level of tuning for the refinement schedule is the same as all other schedules. We do not sweep τ , the same value is used for all experiments and visualizations.
> * Our method does require two training runs in the case when you have no existing gradient norm logs to base the schedule on. We are actively researching methods that provide similar adaptivity in a more online fashion, and we believe such approaches are an exciting research frontier, there is currently no known adaptive methods that work well for single runs.
>
>
> We see that your opinion of our work is overall favorable, and we hope that with these revisions you will revise your score. We think our work has the potential for immediate impact with the community, given it is simple to use and provides substantial improvements in practice. Thank you!

---

### Official Review · Reviewer_NM9Z · 2023-11-01

**Soundness:** 3 good
**Presentation:** 3 good
**Contribution:** 2 fair
**Rating:** 5
**Confidence:** 4

**Summary:**

This paper investigates the learning rate schedules for SGD and study the convergence of the last iteration. The proposed method achieves a problem-adaptive learning rate schedule without using the crude constant bounds on the gradient norms, and proved to be effective via numerical experiments.

**Strengths:**

1. This paper investigates the last iteration convergence for SGD and shows that the best choice is the linear decay schedule. This finding is also validated by solid numerical experiments. Overall, the authors present interesting results in this paper.
2. This paper is well-organized and easy for readers to follow. The proofs in the paper seem correct to me.

**Weaknesses:**

1. The contribution of this paper is limited. The $\frac{1}{\sqrt{T}}$ convergence is not new, and it would be better if the authors could highlight the difference (novelty) and the challenge in the analysis of this paper.

2. The statement that the best strategy is the linear decay schedule seems not to be well supported. Although in the author's analysis, the proposed method can be reduced to the decay, it cannot theoretically prove that it is better than other methods.

**Questions:**

See Weaknesses Part

---

> ### Author Response · Authors · 2023-11-13
> **Response**
>
> Thank you for the appreciative comments, we are grateful that you find our work easy to follow, given the complexity of the subject matter!
>
> # Contribution
> Our contribution has both a theoretical and practical aspect.
>
> ### Theory
> In terms of the theory, as you state, the convergence rate of 1/sqrt(T) for the convex Lipschitz complexity class is well known and achieved by SGD with a linear decay schedule. This rate is worst-case optimal, as simple problems can be constructed where no method can do better.
>
> Our contribution is a learning rate schedule that is *adaptive*, in that it takes advantage of problem structure when it is available, while still working as well as a non-adaptive schedule on worst case problems. In this case, the linear-decay schedule is optimal for the worst-case, where the gradient norms are all of approximately equal magnitude. In the common case where gradient norms increase or decrease over time, our method can be significantly faster! Comparing upper bounds, we find that our upper bound is tighter than the linear decay bound by a range from 9 to 1.5 times for the deep learning problems considered, a very significant practical difference.
>
> Our upper bound is always at least as tight as that for the linear decay schedule, but as you state, we do not prove that refinement is actually superior, just that it has a tighter upper bound in some situations. We think this is still a very significant contribution, as no scheduling method was known prior to our work that could achieve a better upper bound!
>
> ### Practice
> Our practical contributions are also very significant by themselves. We provide the largest ever comparison of learning rate scheduling methods to appear in the literature. This is a useful reference for practitioners looking to choose a learning rate schedule to use, as our comparison covers a very large and diverse set of problems.
>
> In terms of our refinement method, we are able to show clear improvement on the majority of problems over the current state-of-the-art schedule in use, a major achievement. There are no existing efficient adaptive methods for learning rate scheduling that have shown any improvement over common default schedules.
>
> ### Linear decay schedule
> Our statement that the linear decay schedule is the best non-adaptive schedule was intended to be an empirical observation based on our experiments. We are sorry for the confusion there, we will clarify this in the camera ready.
>
> # Overall comments
> We hope that our discussion of the weaknesses that you point out will go some way in swaying you regarding the significance of our work. We believe our method provides a very valuable contribution to the literature even given the limitations you mention.

---

### Official Review · Reviewer_BqWo · 2023-11-01

**Soundness:** 4 excellent
**Presentation:** 3 good
**Contribution:** 3 good
**Rating:** 5
**Confidence:** 3

**Summary:**

This paper delivers an advanced examination of learning rate schedules in the context of stochastic gradient descent. The authors depart from conventional techniques and introduce a novel approach to derive a problem-adaptive learning rate schedule. Furthermore, the paper conducts an extensive evaluation of learning rate schedules, establishing that their schedule refinement technique yields further enhancements.

**Strengths:**

1. The authors offer a comprehensive theoretical analysis of the problem-adaptive learning rate schedule, providing detailed insights into its workings.

2. The authors conduct an extensive evaluation of learning rate schedules, directly comparing classical and modern schedules. Their findings reveal a clear hierarchy among these schedules, offering valuable insights into their relative effectiveness.

**Weaknesses:**

1. One limitation of this paper is the absence of theoretical analysis in non-convex settings, which is particularly relevant in deep learning problems.

2. From a theoretical perspective, it is not evident how the proposed method outperforms other classical learning rate schedules. Clarifying the advantages of this approach in comparison to traditional methods is essential for a comprehensive understanding of its efficacy.

**Questions:**

1.Regarding the deep learning experiments:
1.1. It's not explicitly mentioned whether GPT, RoBERTa, and ViT train from scratch. Additional details on the training process would be beneficial.
1.2. Table 4 indicates that after an extended training period (epochs=30), the cosine learning rate schedule yields superior results compared to the linear decay schedule. It's a valid inquiry to explore whether, with even longer training, the same pattern might emerge for GPT and RoBERTa training—i.e., whether cosine decay becomes more advantageous.
1.3. It would be valuable to clarify whether the learning rate scheduling method labeled 'Cosine' refers to the classic cosine decay or cosineannealing learning rate schedule.

2.Adagrad is known to perform well under convex settings and has strong theoretical support in such scenarios. . However, it is crucial to investigate whether this paper demonstrates that the proposed method surpasses Adagrad, both theoretically and empirically. Further elaboration, as well as empirical comparisons, would be required to draw a definitive conclusion in this regard.

3.Can Theorem 8 be applied to the LAMB optimizer?

---

> ### Author Response · Authors · 2023-11-13
> **Response**
>
> Thank you for the constructive feedback. You raise some excellent points which we would like to address one-by-one:
>
> * For the GPT, RoBERTa and ViT problems, our training runs are not fine-tuning (i.e., they are trained from scratch). Sorry for omitting this, we will include this information in the camera ready. Our experiments are significantly larger scale than any prior work on scheduling. We have endeavored to provide experiments truly comprehensive enough to draw concrete conclusions.
> * Good Point! Any training run that significantly overfits the data would potentially see the same problem as observed for CIFAR, so we would expect that this could occur with GPT and RoBERTa training if they were trained for a much longer time. It's much harder for the gradient norm to completely collapse on those problems but it could occur, particularly if training on small datasets. We will add a note on this to the paper. Thank you for pointing this out!
> We think that this issue can be mitigated by using the gradient norms on held out data rather than the training data, however that obviously complicates the implementation of the method. We are currently investigating this direction.
> * We debated including AdaGrad comparison in this paper. For the majority of problems considered, AdaGrad does significantly worse than Adam in practice, so it's hard to do a fair comparison, as Adam+Refinement is much better than AdaGrad by itself. AdaGrad can be combined with refinement, and could potentially be another interesting baseline. We can include a comparison in the Appendix if you think it would be useful, please let us know!
> * Yes, our approach can be combined with the LAMB optimizer, it just requires deriving the formula for the weighted norm to measure gradients in. A layer-wise weighted L1 norm will likely be a good approximation for practical use.
>
> ## Weakness
> 1) For the stochastic non-convex setting, there is existing theory (https://arxiv.org/pdf/1910.01845.pdf) that shows that any last-iterate rate is actually impossible (Sec. 3.1). It's an interesting research question to determine what additional assumptions are needed in the non-convex setting to “bridge-the-gap”, as seemingly the convex case is much more representative of the scheduling behavior in practice for deep learning problems, and as yet nobody understands why. We are hoping to make progress on this but it's a difficult problem.
> 2) We believe our theoretical rate for the refinement method provides a strong indication of when our method is expected to outperform the linear decay baseline. In any situation when the gradient norm sequence is not flat, our bound is tighter than the linear decay method's bound, by a factor that we can concretely compute after observing the gradient norm sequence. Our bound is always at least as good as the linear decay bound.
>
> We ran these numbers, and it ranges from a factor of 9 to a factor of 1.5 for the deep learning problems considered. This doesn't guarantee an improvement as we are just comparing two upper bounds, but we believe it's indicative of when our method has an advantage.
> This guarantee can be made more concrete by considering classes of problems where the gradient norms change over time in a structured way, such as strongly-convex problems, or problems satisfying relative smoothness conditions. We intend to pursue this direction in follow-up work.
> ## Overall
> We appreciate the very positive view you take of our paper. We hope that we have addressed your questions above. We believe our paper has the potential to be very impactful, as we have a method that was derived from a strong theoretical foundation, and also works extremely well in practice, out-performing strong baselines in the majority of problems. We believe this is the first practical method developed for learning rate scheduling for deep learning problems. We hope that you will consider raising your score to an 8 in light of our comments. Thank you!

---

### Official Review · Reviewer_eGse · 2023-11-07

**Soundness:** 4 excellent
**Presentation:** 3 good
**Contribution:** 4 excellent
**Rating:** 8
**Confidence:** 4

**Summary:**

This paper presents a refined study of learning rate scheduling for last iterate convergence of stochastic gradient methods. This automatically yields warmup and annealing schedules, and predicts a linear decay when gradient information is unavailable. The paper also presents an extension to co-ordinate wise methods and supplements this with empirical studies on many deep learning benchmarks including LLMs. Interestingly, this paper presents a learning rate scheduling scheme for any no-regret learning method into one that offers a last iterate convergence guarantee.

**Strengths:**

- A very refined characterization of learning rate scheduling that captures various nuances relating to warmup, annealing etc. It recovers several practically effective heuristics that have lacked theoretical support in prior works.
- A reasonably thorough treatment of empirical benchmarking with many deep learning problems of interest.

**Weaknesses:**

The paper's writing can be made clearer about notions of anytime optimality versus developing schemes that work assuming a known end time (as is done in this paper), and what are the challenges in developing an algorithm for the unknown end time case?

**Questions:**

- What sequence of learning rates obtain optimal rates in terms of gradient norm? Can this potentially address the limitation mentioned at the end of this paper?
- Can the authors comment on whether one can utilize the doubling trick (Hazan and Kale 2014) to make progress on the unknown end time case?
- Another popular heuristic in practice (and in theory) is that of batch size doubling. Can the authors comment on how (or whether) these results can be connected with how to set batch sizes in practice?

---

> ### Author Response · Authors · 2023-11-13
> **Response**
>
> Thanks for the valuable feedback! To answer your points:
> * Any-time convergence is, we believe, the next big unsolved problem in learning rate scheduling. There are no existing approaches that perform well in practice. We are actively researching techniques that bridge this performance gap. We will add more discussion of this to the paper following your suggestion.
> * If the learning rate was set using the gradient norms computed on held-out data, rather than the training data, this will avoid the issues we see on problems that contain overfitting. We are actively researching this and other solutions.
> * This is a good idea! The use of restarting with doubling could be used to give any-time rates. There is usually some performance overhead to doubling methods in practice, but it's definitely worth trying out. Thank you for the suggestion.
> * This is an interesting question. It would be worth looking at how the gradient norm graphs change as the batch size is increased. Without some experimental guidance on what occurs, it's hard to guess what the behavior would be. If we make a theoretical assumption such as the additive or multiplicative noise assumptions, then we could analyze the behavior very concretely.
>
> Please let us know if you have any other questions.

---

### Author Response · Authors · 2023-11-20
**Discussion ends soon!**

We hope that we have answered all questions posed by reviewers to their satisfaction. We are here and ready to answer any further questions during the discussion period, which ends in 2 days.

We see that there is a strong positive reception of our work, with some misgivings about the generality of our theory. Since theoretical results of the kind we establish are impossible in the non-convex setting (see our rebuttals), our results are in one of the most general settings possible, and we hope that reviewers will consider this in their ratings.

In terms of our contributions, we would like to highlight some aspects of our work:
 - The first approach to adaptive learning rate scheduling that out-performs existing SOTA schedules across a large suite of problems.
 - The largest ever comparison of learning rate scheduling approaches.
 - Our method is backed by novel theory - it arises "theory-first" based on a derivation of the optimal regret weighting. This theory is interesting by its self - we establish that for any sequence of regret weights, there is a corresponding closed-form learning rate schedule that optimizes a tight upper bound on the regret.
 - Extreme simplicity - our method can be implemented in just a few lines of code.

---

### Meta-Review · Area_Chair_eT96 · 2023-12-07

**Metareview:**

In this paper, the authors propose to investigate the learning rate schedule of Stochastic Gradient Descent (SGD) and its convergence in a convex setting. The paper introduces a practical framework for determining learning rates. However, the theoretical contribution, claimed as a main novelty, is deemed impractical in real-world scenarios. The majority of reviewers feel the paper is not ready for publication and recommend rejection.

The usefulness of Theorem 3 in practice is questioned since it requires the weight parameter $w_t$ at iteration $t$ to depend on stochastic gradients $g_1, \ldots, g_T$, which are not available to the algorithm. Additionally, the proof of Theorem 3 relies on the independence between $w_t$ and $g_t," which contradicts the proposed schedule. Hence, it is doubtful whether the derived convergence bound in the theorems holds in realistic settings.

The paper only explores the convex setting, and the convergence rate of $1/\sqrt{T}$ is not novel in the literature. The claimed novelty revolves around the adaptive learning rate schedule in their algorithm. However, as discussed earlier, this learning rate schedule faces issues and is not implementable as it depends not only on historical iterates but also on future iterates. Consequently, the theoretical contribution of the paper needs significant improvement.

I encourage the authors to further investigate the validity of the theory and provide more justification for their statement on the optimality of the linear decay schedule.

**Justification For Why Not Higher Score:**

The theoretical contribution of this paper relies on overly idealistic assumptions, resulting in a vacuous claim about the improvement of convergence rates.

**Justification For Why Not Lower Score:**

N/A

---

### Decision · Program_Chairs · 2024-01-16

Reject